# How Powerful Are LLMs in Generating Formal Program Specifications?

**Fanpeng Yang** [* 1 2] **Xing Li** [* 1 2] **Shuling Wang** [1 2] **Jie An** [1 2] **Zeyu Sun** [1] **Shenghua Feng** [1] **Wenhan Wang** [1] **Weiyi Wang** [1] **Naijun Zhan** [3 4] **Fanjiang Xu** [1 2]

## Abstract

Formal verification provides strong guarantees of software correctness, but its adoption is limited by the high cost of writing precise formal specifications. While recent large language models (LLMs) have shown strong capabilities in theorem proving and verified code generation, their true ability to generate program specifications remains unclear. Existing evaluations require either verifying implementation conformance or proving semantic equivalence between specifications, both of which are formidably difficult and may conflate proof difficulty with specification quality. To address this problem, we introduce COINS, a Rocq based evaluation framework that assesses specification quality by instantiating specifications under evaluation on trusted test cases and generating concrete proof obligations. This design aligns with the asymmetric nature of formal reasoning, where successful proofs provide reliable evidence while proof failures are inherently ambiguous. Using COINS, we conduct a large scale study on HumanEval with a curated set of human written Rocq specifications. Our results show that specification generation remains a formidable challenge, and that verification complexity can obscure genuine differences in specification quality. Overall, we find that accurate specification evaluation, rather than model scaling alone, is central to understanding the power of LLMs for specification synthesis, and that test case based formal reasoning offers a more faithful and discriminative measure of progress.

---

[*]Equal contribution [1]Institute of Software, Chinese Academy of Sciences, Beijing, China [2]University of the Chinese Academy of Sciences, Beijing, China [3]School of Computer Science & Key Lab. of High Confidence Software Technologies of Ministry of Education, Peking University, Beijing, China [4]Zhongguancun Lab., Beijing, China. Correspondence to: Jie An <anjie@iscas.ac.cn>, Zeyu Sun <zuyu.zys@gmail.com>, Shenghua Feng <feng-shenghua@iscas.ac.cn>.

*Proceedings of the 43rd International Conference on Machine Learning*, Seoul, South Korea. PMLR 306, 2026. Copyright 2026 by the author(s).

## 1. Introduction

Formal verification provides the strongest guarantees of software correctness by enabling machine-checkable proofs that an implementation satisfies its intended behavior. Interactive theorem provers (ITPs) such as Rocq (Huet et al., 1997), Lean (De Moura et al., 2015), and Isabelle (Paulson, 1994) have been successfully applied to the verification of large-scale, safety-critical systems, including certified compilers and operating system kernels. Despite these successes, the practical adoption of formal verification remains fundamentally constrained by the cost of writing formal specifications. Constructing specifications that are both semantically precise and sufficiently restrictive to rule out unintended behaviors requires expert knowledge and substantial manual effort, often exceeding the effort required to verify the implementation itself.

Recent advances in large language models (LLMs) have demonstrated that neural models can meaningfully assist formal reasoning tasks. In particular, LLMs have achieved notable success in formal mathematical theorem proving (Xin et al., 2024; Wang et al., 2024; Xin et al., 2025; Ren et al., 2025; Chen et al., 2025a; Lin et al., 2025; Wang et al., 2025; Hubert et al., 2026), where models generate or complete proofs that are mechanically checked by proof assistants. These results indicate that LLMs can internalize non-trivial logical structure and interact effectively with formal languages. Building on this progress, recent work has begun to transfer these capabilities to program verification, where LLMs are used for tasks such as invariant inference (Kamath et al., 2023; Chakraborty et al., 2023; Pirzada et al., 2024; Cao et al., 2025b), translating natural language into formal specifications (Cosler et al., 2023; Endres et al., 2024; Cao et al., 2025a; Fang et al., 2025), and program synthesis (First et al., 2023; Misu et al., 2024). This progression naturally raises a further question: *how powerful are LLMs at generating formal program specifications?* Unlike theorem proving or implementation synthesis, specification generation requires translating informal intent—expressed through natural language descriptions or reference implementations—into precise logical predicates that characterize program behavior.

A central challenge in answering this question lies in how to

evaluate specification quality at scale. Relying solely on human experts is not only inefficient but also cannot guarantee correctness, as even experts may overlook subtle semantic errors. Most existing work in program verification evaluates specifications indirectly by checking whether a given implementation satisfies them. In these work, specifications take the form of verification objectives, or error-checking conditions, and are designed primarily to discharge a specific verification goal rather than to fully characterize program behavior. While effective for validating implementations against targeted properties or ruling out particular classes of errors, this paradigm provides only a partial view of specification quality: a specification may be sufficient to verify a particular implementation or objective yet remain overly permissive, failing to rule out unintended behaviors or alternative incorrect implementations. In other words, traditional verification pipelines are fundamentally oriented toward establishing *implementation* $\models$ *specification*, but offer limited ability to assess whether a specification itself precisely captures the intended semantics of the program.

Recent benchmarks have attempted to strengthen specification evaluation by imposing stricter correctness criteria. CLEVER (Thakur et al., 2025), for example, proposes an end-to-end benchmark that requires models to generate specifications together with implementations and proofs of equivalence against a claimed ground-truth specification, all expressed within a theorem-proving language. While conceptually appealing, this design introduces substantial practical challenges.

First, while equivalence proofs can in principle avoid over-approximation, they reintroduce dependence on human-curated ground-truth specifications—resources that demand substantial effort to produce and may themselves introduce bias or inconsistency. Second, even verifying that a generated implementation satisfies a generated specification in a fully formal setting is already highly challenging. Requiring, in addition, full equivalence proofs between non-trivial specifications further increases the proof burden, resulting in very low success rates even for state-of-the-art models. Third, because specification generation, implementation generation, and formal verification are tightly coupled in a single pipeline, failures are difficult to diagnose: it is often unclear whether an unsuccessful result reflects an incorrect specification, an incorrect implementation, or limitations of the proving process itself. As a result, although CLEVER enforces a strong notion of correctness, its evaluation signal is sparse and hard to interpret, limiting its usefulness for assessing specification quality in isolation.

These limitations suggest the need for an evaluation methodology that can assess specification quality within a tractable and interpretable formal reasoning framework. In this work, we argue that specification evaluation should focus on what

can be reliably established through formal proofs rather than relying solely on all-or-nothing semantic equivalence checking.

Motivated by these considerations, we propose COINS (**CO**q-based **IN**stantiated **S**pecification evaluation), an evaluation framework for assessing LLM-generated formal specifications using the Rocq proof assistant. The name reflects the core idea of our approach: specifications are evaluated by instantiating them on concrete behaviors and reasoning about the resulting proof obligations in Rocq. We adopt Rocq because it is one of the most widely used proof assistants for program verification (Leroy et al., 2016; Cao et al., 2018; Zhou et al., 2024; Wu et al., 2025), with a long history of successful applications in large-scale verified systems such as CompCert (Leroy et al., 2016). We chose Rocq over Lean because Rocq has a longer and more established track record in program verification, making it a more natural fit for evaluating program specifications.

Using COINS, we conduct a large-scale empirical study of formal specification generation in Rocq on the HumanEval benchmark (Chen et al., 2021). All 164 ground-truth specifications were manually written and cross-reviewed by researchers with multiple years of theorem-proving experience, requiring several person-weeks of expert effort; to the best of our knowledge, this is the first complete formal specification suite for HumanEval in Rocq. We use this suite as a reference baseline and evaluate a diverse set of state-of-the-art LLMs under a unified evaluation pipeline. Our evaluation clearly differentiates the specification generation capabilities of different models, revealing substantial performance gaps: COINS scores range from 28.05% for Gemini 3 Pro Preview to 1.22% for DeepSeek-V3.1. While frontier models can approach human-level performance on some tasks, generating tight formal specifications remains a significant challenge.

Our main contributions are as follows:

- We identify a fundamental limitation of existing program verification pipelines: while effective at checking whether implementations satisfy specifications, they provide limited ability to assess specification quality itself.
- We propose COINS, a principled evaluation framework for specification quality based on formally provable behavior in Rocq.
- We construct a curated dataset of human-written Rocq specifications for all 164 HumanEval problems and perform a comprehensive empirical evaluation of state-of-the-art LLMs.

## 2. Program Specification

We first clarify what we mean by a *formal program specification* and why evaluating it differs from evaluating implementations.

Consider the `sort_third` problem from HumanEval: given a list, return a modified list in which elements at indices divisible by three are sorted, while all other elements remain in their original positions. Correct implementations may use different algorithms, data structures, or iteration styles.

A specification abstracts away all such details. It is a logical proposition (`Prop`) over *only* the input and output, capturing the relationship any correct implementation must satisfy. It should be independent of any particular implementation and admit no over-approximation on the input–output relation.

Concretely, the specifications we consider consist of two parts: a *precondition* characterizing properties the input must satisfy, and a *postcondition* characterizing the relation the input and output must jointly satisfy. By design, such specifications are agnostic to both the implementation language and the implementation itself, making no statement about runtime exceptions, memory layout, or other operational artifacts. While such properties may legitimately belong in language-specific specifications (e.g., a C-level memory contract), these lie outside the scope of this work, which targets language- and implementation-agnostic specifications of functional behaviour.

For `sort_third`, this can be written in Rocq as:

```
Definition sort_third_spec
  (input output : list Z) : Prop :=
  Permutation input output /\
  (forall (i : nat),
    (i < length input)%nat ->
    (i mod 3 <> 0) ->
    nth i output 0%Z = nth i input 0%Z) /\
  (forall (i j : nat),
    i < length output /\
    j < length output /\
    i mod 3 = 0 /\
    j mod 3 = 0 /\
    i < j ->
    (nth i output 0 <= nth j output 0)%Z).
```

This specification states three conditions: the output is a permutation of the input, non-divisible-by-three positions are preserved, and divisible-by-three positions are sorted in non-decreasing order. Any correct implementation satisfies this specification, and any incorrect one violates at least one condition.

This differs from program translation: mechanically translating Python into Rocq would encode algorithm-specific details and remain tied to one implementation. A specification, by contrast, captures only what the user cares about—

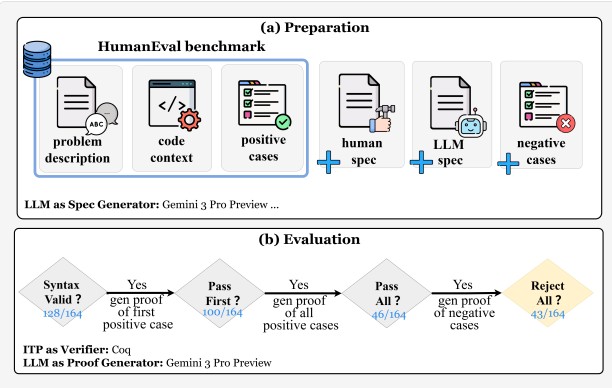

*Figure 1.* **The COINS evaluation framework.** (a) Preparation: inputs from HumanEval and generated specifications. (b) Evaluation: multi-stage filtering pipeline. Annotated numbers show filtering results for Gemini 3 Pro Preview (164 → 43 candidates).

for algorithmic problems, functional correctness rather than *how* the result was computed.

Under this definition, a specification is not a problem with a single canonical answer. For functional correctness, test cases provide the most natural formalization of user concerns: positive cases encode accepted behaviors and negative cases encode rejected ones, and proving on concrete test cases is substantially easier than proving full semantic equivalence, allowing us to isolate specification quality from prover capability.

This observation directly motivates COINS, which evaluates a specification by instantiating it on concrete test cases and checking whether the resulting proof obligations can be discharged. For the positive test case (`[2;1;3;7;8;9;10]`, `[2;1;3;7;8;9;10]`), COINS asks the LLM to prove:

```
Example sort_third_pos :
  sort_third_spec [2;1;3;7;8;9;10]
                  [2;1;3;7;8;9;10].
Proof. unfold sort_third_spec. ... Qed.
```

For a mutated negative case such as (`[2;1;3;7;8;9;10]`, `[7;1;3;2;8;9;10]`), the specification should *not* be provable; if the LLM constructs a proof, the specification is overly permissive and fails. The object evaluated is always the specification itself, not the implementation. We describe the full evaluation pipeline below.

## 3. Evaluation Framework

To evaluate program specifications written in Rocq, we instantiate each specification with concrete input-output values from the test cases and discharge the induced verification obligations. A specification is deemed to accept a test case if and only if the corresponding verification obligation

can be successfully constructed and verified by the Rocq proof assistant. We demonstrate our approach using the HumanEval benchmark.

Figure 1 illustrates our evaluation pipeline, organized into two phases. In the **Preparation Phase** (Figure 1a), we manually craft reference Rocq specifications for HumanEval problems, prompt LLMs to produce specifications following the same signature, and generate negative cases through mutation testing to supplement the original positive test cases. In the **Evaluation Phase** (Figure 1b), we progressively filter specifications through a multi-stage pipeline—syntactic validation, positive test acceptance, negative-case rejection, and equivalence verification—by constructing acceptance proofs at each stage.

### 3.1. Preparation Phase

The preparation phase establishes the evaluation infrastructure through three key components: (1) human specifications that serve as the ground-truth, (2) LLM prompts for eliciting LLM-generated specifications, and (3) test suites combining both positive and negative cases to assess specification behavior.

**Human Specifications.** We manually constructed ground-truth Rocq specifications for all 164 problems in the HumanEval dataset. Each specification takes the form of a `Prop` that defines a logical predicate over input and output, following a unified signature. To ensure semantic correctness, each specification underwent rigorous review by experts in formal verification. These human-authored specifications serve a critical baseline in our framework.

**LLM Prompting Strategy.** We design prompts that provide LLMs with the natural language description and reference implementation from each HumanEval problem, along with explicit instructions to generate Rocq specifications adhering to the same unified signature as our reference specifications. This signature constraint ensures structural consistency between LLM-generated and human-authored specifications, enabling direct comparison in subsequent evaluation stages. The actual specification generation occurs in the evaluation phase, where we query multiple LLMs using these prepared prompts.

**Test Suite Construction.** To evaluate specification behavior, we construct a dual test suite comprising both positive and negative test cases, providing complementary coverage of correct and incorrect program behaviors.

*Positive Test Cases.* We leverage positive test cases directly from the HumanEval+ benchmark, which augments each HumanEval problem with extensive automatically-generated input-output pairs, averaging 755.98 positive test cases per problem. These cases represent intended program behaviors and provide broad coverage of the specification's

acceptance criteria.

*Negative Test Cases.* To identify overly permissive specifications, we generate negative test cases via mutation testing on the canonical HumanEval implementations. We apply operator-level mutations and collect an input-output pair as a negative case whenever a mutant's output diverges from the ground-truth behavior. This procedure produces 10 negative test cases per problem, for a total of 1,640 negative cases across all 164 HumanEval tasks. Detailed mutation operators and generation procedures are provided in the Appendix A.

### 3.2. Evaluation Phase

In the evaluation phase, we first prompt LLMs to generate Rocq specifications using the prepared prompts from the preparation phase. The generated specifications then undergo a multi-stage filtering pipeline (Figure 1b), where each stage incrementally accumulates evidence about specification correctness and precision. The detailed design principles of this pipeline, along with an analysis of each stage, are provided in Appendix B.

**Syntactic Validation (SYNTAX).** We first check whether each generated specification is syntactically well-formed and successfully compiled in Rocq. Specifications that fail this stage are discarded without further semantic evaluation.

**First-Case Acceptance ($\text{PASS}_{\text{first}}$).** For syntactically valid specifications, we attempt to discharge the proof obligation corresponding to the first positive test case. This early filtering step reduces computational overhead in later stages.

**Full Positive Test Acceptance ($\text{PASS}_{\text{all}}$).** Specifications that pass the first case are evaluated against all positive test cases. We reuse the proof constructed for the first case as a reference, as successful proofs often share structural patterns across test instances. Failure to discharge some test obligations is treated as *inconclusive*, as it may arise either from an incorrect specification or from limitations in automated proof synthesis.

**Negative Test Rejection ($\text{REJECT}_{\text{all}}$).** Passing all positive test cases alone is insufficient to rule out overly permissive specifications. We evaluate specifications against automatically generated negative test cases. For each negative case, we prompt the LLM to judge whether the specification accepts it; if so, the LLM attempts to construct a formal acceptance proof, where success indicates the specification is overly permissive. If the LLM judges rejection, we skip the formal proof as proving rejection can be harder. A specification passes $\text{REJECT}_{\text{all}}$ only if no negative example can be proven acceptable. $\text{REJECT}_{\text{all}}$ serves as our core metric: only specifications satisfying both $\text{PASS}_{\text{all}}$ and $\text{REJECT}_{\text{all}}$ are designated as **candidate specifications**.

In practice, candidate specifications serve as a reliable starting point for full formal verification, where correctness established on test cases is generalized to all inputs satisfying the precondition. This avoids the costly scenario of attempting complete formal proofs on flawed specifications.

# 4. Experiments

We evaluate the ability of LLMs to generate formal specifications. While recent work has explored LLM-assisted verification and proof synthesis, the quality of LLM-generated specifications themselves remains largely unexamined. This gap is nontrivial: when verification fails, it is often unclear whether the cause lies in an incorrect specification or in limited proof capability. To address this challenge, we formulate four research questions (RQs), each supported by a targeted experimental design within the COINS framework shown in Section 3.

Our study is organized around four research questions. RQ1 aims to identify the most reliable verifier using human-crafted specifications, thereby anchoring one side of the evaluation duality; building on this, RQ2 evaluates LLM-generated specifications through our multi-stage pipeline to reveal where generation fails, with the two together characterizing the intrinsic difficulty of specification generation. RQ3 addresses the interference from verification difficulty by disentangling errors caused by weak specifications from those due to insufficient proof power. Finally, RQ4 investigates the impact of specification style, examining how the use of executable `Fixpoint` definitions versus relational `Inductive` predicates affects clarity, compositionality, and verifiability.

Additional experimental results and analyses, including the human-vs-LLM specification comparison, error analysis, case studies, and comparisons with alternative evaluation metrics, are deferred to Appendices E, F, G, H, I, and J.

**Model Selection.** To ensure a comprehensive evaluation, we select six large language models from four leading organizations. Our selection includes frontier models from Q4 2025: **GPT-5** (OpenAI), **Claude 4.5 Opus** (Anthropic), and **Gemini 3 Pro Preview** (Google), as well as widely adopted earlier models: **GPT-4o**, **Claude 3.7 Sonnet**. We also include **DeepSeek-V3.1**, a state-of-the-art, open-source model from DeepSeek. This diversity enables evaluation across both capability tiers and organizational implementations.

**Experimental Configuration.** All models are accessed via their official APIs under a unified evaluation protocol. For each proof generation task, models are allowed up to three attempts, with compiler or verifier error feedback provided between attempts. Detailed model configurations and the full prompt templates used in all experiments are provided in Appendix C.

*Table 1.* Verification Performance Across Different Models (n=164). *Models ordered chronologically by release date. Best performing model highlighted. Higher values indicate better performance.*

| Verify Model | PASS$_{first}$ | PASS$_{all}$ |
|---|---|---|
| GPT-4o | 37.20% (61) | 7.32% (12) |
| Claude 3.7 Sonnet | 30.49% (50) | 3.66% (6) |
| GPT-5 | 60.98% (100) | 16.64% (27) |
| DeepSeek-V3.1 | 46.95% (77) | 0.00% (0) |
| Claude 4.5 Opus | 72.56% (119) | 19.51% (32) |
| Gemini 3 Pro Preview | **74.39%** (122) | **29.88%** (49) |

## 4.1. RQ1: Which LLM Serves as the Most Reliable Verifier?

A core challenge in assessing specification quality is its inherent dependence on verification. In our early experiments, we attempted to verify specifications by proving that canonical implementations conform to them—the standard approach in formal verification. However, we found that such proofs are extremely complex and resist automation even when the specification is semantically correct. This difficulty is not unique to our setting: even in established benchmarks (Thakur et al., 2025; Ye et al., 2026), ground-truth specifications are often validated only informally, rather than via rigorous formal proofs against implementations. We highlight this as a key observation:

> **Evaluation Duality.** When a specification fails to be verified, it is unclear whether the fault lies in the specification itself or in the prover. Stronger provers more accurately expose specification correctness, while more correct specifications more reliably assess prover capability.

To break this cyclic dependency and minimize evaluation uncertainty, we anchor one side of the duality with our manually crafted specifications. While we cannot claim these as absolute ground truth, they have undergone careful human review and we believe them to be of higher quality than LLM-generated alternatives. Using these reference specifications, we evaluate multiple LLMs by measuring their PASS$_{all}$ rate on the associated proof objectives. The best-performing model is then adopted as the verifier for all subsequent experiments, providing the most reliable assessment of generated specification quality.

Based on the evaluation shown in Table 1, we identify Gemini 3 Pro Preview as the strongest overall performer in proof synthesis. Consequently, we select Gemini 3 Pro Preview as the primary proof engine for subsequent experiments. However, even Gemini 3 Pro Preview's performance remains far from ideal—achieving only 29.88% PASS$_{all}$ on human-

crafted specifications highlights the substantial gap between current LLM capabilities and reliable automated theorem proving.

## 4.2. RQ2: What Observations Can be Derived from the Evaluation Results?

Having identified Gemini 3 Pro Preview as the strongest verifier in RQ1 (Sec 4.1), we employ it as the primary verifier to evaluate specifications generated by all six models on COINS. Table 2 presents the results, where $\text{REJECT}_{all}$ serves as our core metric for identifying candidate specifications.

**Syntax Correctness Is Still Challenging.** Generating syntactically valid Rocq specifications remains a fundamental challenge for most models. While Gemini 3 Pro Preview achieves 78.05% syntax correctness and GPT-5 reaches 67.07%, earlier-generation models struggle significantly—DeepSeek-V3.1 produces only 7.93% valid specifications, and GPT-4o achieves just 14.63%. This syntactic barrier filters out the majority of attempts before semantic evaluation can even begin. We provide a detailed analysis of syntax error causes in Appendix G.

**Substantial Performance Gap.** Significant disparities exist between frontier and earlier-generation models. Syntax correctness ranges from 78.05% (Gemini 3 Pro Proview) to 7.93% (DeepSeek-V3.1), with this gap amplifying at later stages—GPT-4o achieves only 4.27% $\text{REJECT}_{all}$ versus 26.22% for Gemini 3 Pro Preview.

**Limited Self-Verification Bias.** Self-verification experiments confirm Gemini 3 Pro Preview's lead is not due to evaluating its own outputs: strong models achieve comparable or lower $\text{PASS}_{all}$ under self-verification (e.g., Claude 4.5 Opus: 10.98% vs. 14.63%).

**Precision Emerges from Coverage.** The close alignment between $\text{REJECT}_{all}$ and $\text{PASS}_{all}$ indicates that specifications passing all test cases are typically precise enough to reject incorrect implementations. This can be attributed to two factors: our diverse test suite effectively constrains the semantic space, and for relatively simple programs, there is little room for over-approximation in specification semantics, which aligns with the findings in (Le-Cong et al., 2025).

**Comprehensive Test Suites Are Essential.** The sharp drop from $\text{PASS}_{first}$ to $\text{PASS}_{all}$ across all models (e.g., Gemini 3 Pro Preview: 60.98%→28.05%) confirms that individual test cases are insufficient—multiple tests collectively constrain specification semantics and distinguish genuine correctness from accidental success.

**Syntax Is the First Bottleneck.** For relatively simple algorithmic tasks such as HumanEval, syntactically valid Rocq specifications are often semantically precise under our test-based evaluation. The dominant failure mode is not that models systematically choose the wrong behavioral intent, but that they hallucinate invalid Rocq syntax or ill-typed formal constructs while trying to express program behavior.

## 4.3. RQ3: Why Is Specification Quality Difficult to Evaluate?

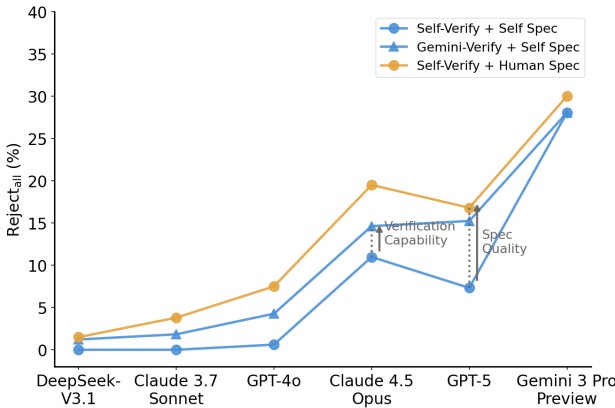

*Figure 2.* Ablation study disentangling specification quality and verification capability. Compared to the *Self-Verify + Self Spec* baseline, using human-written specifications improves performance by +5.01%, while replacing self-verification with a stronger verifier improves performance by +3.05%. Both factors significantly impact performance.

**Verification Complexity Trade-off.** The harder the verification task, the stronger the prover required, and the more difficult it becomes to accurately assess specification quality. Even when the verification objective perfectly captures specification correctness, overly complex proof obligations risk systematic underestimation, rendering such benchmarks unreliable metrics for specification generation.

Compared to CLEVER's end-to-end approach, which achieves only 0.621% correctness and provides virtually no discriminative power across models, COINS successfully differentiates specification generation capabilities among all six models. As shown in Table 2, $\text{REJECT}_{all}$ rates range from 1.22% (DeepSeek-V3.1) to 28.05% (Gemini 3 Pro Preview), demonstrating meaningful variance across the model spectrum.

However, even the task of verifying test case acceptance proves highly challenging. Table 1 and Table 2 reveal that while models can often verify individual test cases, consistent verification across all cases remains elusive. This makes it difficult to determine whether failures stem from proof synthesis limitations or genuine specification defects.

To disentangle these factors, we conduct an ablation study

*Table 2.* Specification Generation Results (n=164). *Syntax denotes syntactically valid specifications. Higher values indicate better performance.*

| Generate Model | Verify Model | Syntax | PASS$_{first}$ | PASS$_{all}$ | REJECT$_{all}$ |
|---|---|---|---|---|---|
| GPT-4o | Gemini 3 Pro Preview | 14.63% (24) | 12.20% (20) | 4.27% (7) | 4.27% (7) |
| | GPT-4o | | 0.61% (1) | 0.61% (1) | – |
| DeepSeek-V3.1 | Gemini 3 Pro Preview | 7.93% (13) | 4.27% (7) | 1.22% (2) | 1.22% (2) |
| | DeepSeek-V3.1 | | 1.22% (2) | 0.00% (0) | – |
| Claude 3.7 Sonnet | Gemini 3 Pro Preview | 19.51% (32) | 15.24% (25) | 1.83% (3) | 1.83% (3) |
| | Claude 3.7 Sonnet | | 4.27% (7) | 0.00% (0) | – |
| Claude 4.5 Opus | Gemini 3 Pro Preview | 59.76% (98) | 45.12% (74) | 14.63% (24) | 14.63% (24) |
| | Claude 4.5 Opus | | 50.61% (83) | 10.98% (18) | – |
| GPT-5 | Gemini 3 Pro Preview | 67.07% (110) | 60.98% (100) | 15.24% (25) | 15.24% (25) |
| | GPT-5 | | 28.66% (47) | 7.32% (12) | – |
| Gemini 3 Pro Preview | Gemini 3 Pro Preview | 78.05% (128) | 60.98% (100) | 28.05% (46) | 28.05% (46) |

with two controlled interventions (Figure 2). We start from a baseline where each model verifies its own generated specifications (*Self-Verify + Self Spec*). We then separately improve either verification capability by replacing self-verification with Gemini 3 Pro Preview (*Gemini-Verify + Self Spec*), or specification quality by substituting LLM-generated specifications with human-written references (*Self-Verify + Human Spec*).

Both factors significantly affect performance. Improving specification quality yields an average gain of $+5.01\%$, while stronger verification contributes $+3.05\%$. Although the former has a larger impact, verification capability remains a substantial factor even for test case acceptance.

### 4.4. RQ4: Rethinking Executability: Should Specifications Be Purely Relational?

When writing program specifications, handling inductive structures is inevitable. In imperative programming languages, this is typically achieved through loops or recursion. In Rocq, there are three primary approaches: (1) built-in higher-order functions or predicates from Rocq's standard library, such as `length` and `fold`; (2) user-defined `Fixpoint` functions, which are custom recursive functions; and (3) user-defined `Inductive` predicates, which are inductively defined logical relations.

The key distinction between the latter two lies in their executability: `Fixpoint` definitions are computationally executable within Rocq, whereas `Inductive` predicates are purely logical relations without computational content. Importantly, these approaches can be freely combined in practice. Consequently, whether a specification is "executable" is not a well-defined mathematical concept—a single specification may contain both executable and non-executable components.

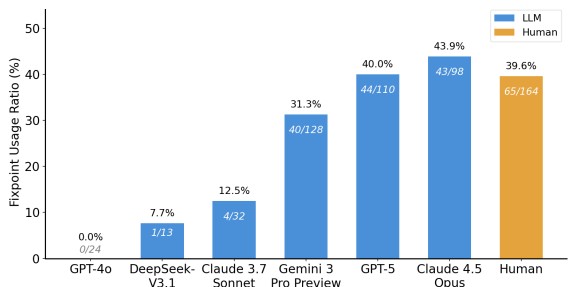

*Figure 3.* Frequency of `Fixpoint` usage across specifications. LLMs exhibit usage ratios (31–44%) comparable to human (39.6%).

Given this observation, the rigid requirement in CLEVER that all manual specifications be non-executable appears overly restrictive in a broader specification-generation setting. This constraint is primarily motivated by CLEVER's design, where implementations are synthesized directly from specifications within Lean, making executability a potential source of specification leakage and trivial solutions. In contrast, our focus is on evaluating specification quality itself rather than end-to-end code synthesis. Accordingly, we focus specifically on the use of `Fixpoint` and provide two versions of our specification suite: one **permitting** `Fixpoint` definitions and one **prohibiting** them. This design enables us to systematically investigate how the availability of `Fixpoint` constructs affects both specification quality and proof difficulty.

Figure 3 presents the frequency of `Fixpoint` usage in specifications. A clear divide emerges: weaker models (GPT-4o, DeepSeek-V3.1, Claude 3.7 Sonnet) rarely employ `Fixpoint` constructs, while stronger models (Gemini 3 Pro Preview, GPT-5, Claude 4.5 Opus) utilize them substantially more often (31–44%). Notably, human experts ex-

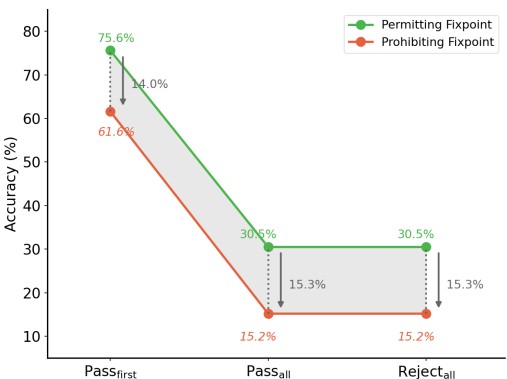

*Figure 4.* Impact of prohibiting `Fixpoint` on human specification quality ($n = 164$). Disallowing recursive constructs leads to substantial performance drops across all metrics.

hibit a similar ratio of approximately 40%, suggesting that state-of-the-art models have developed comparable judgment in determining when recursive structures are necessary for precise formalization.

This pattern indicates that `Fixpoint` plays a crucial role in effective specification design. To validate this hypothesis, we evaluated Gemini 3 Pro Preview on both versions of our human-written specifications. As shown in Figure 4, proofs against specifications prohibiting `Fixpoint` are substantially harder, with a notable drop in the success rate. Furthermore, specifications without `Fixpoint` require significantly more lines of code and exhibit lower readability, as complex recursive logic must be manually unfolded into verbose relational rules. These results lead to the following finding:

> **Executable Components Matter.** Well-designed specifications benefit significantly from executable components. Encapsulating sub-logic within `Fixpoint` definitions effectively reduces both specification complexity and verification difficulty. Pure relational specifications, while theoretically elegant, lead to verbose code and unwieldy proof obligations that challenge even the strongest models.

## 5. Related Work

**LLMs for Interactive Theorem Proving.** The integration of Large Language Models (LLMs) with Interactive Theorem Provers (ITPs) has rapidly evolved from local tactic prediction to the broader challenge of autoformalization. Early Rocq-side benchmarks such as CoqGym (Yang & Deng, 2019) and Proverbot9001 (Sanchez-Stern et al., 2020) enabled supervised learning on large-scale proof traces to suggest individual proof steps. LeanDojo (Yang et al., 2023) then pioneered retrieval-augmented generation for premise selection in Lean, while Baldur (First et al., 2023) and Lemur (Wu et al., 2024) extended this line to whole-proof generation and repair within closed compiler-feedback loops. More recent efforts, including DeepSeek-Prover-v2 (Ren et al., 2025) and APOLLO (Ospanov et al., 2025), introduce reinforcement learning and compiler-guided repair to further boost proving success rates, and NTP4VC (Xu et al., 2026) extends neural theorem proving to industrial verification-condition proving. All of these works focus on finding proofs for *given* specifications. Our work addresses the complementary problem of evaluating the quality of the specifications themselves. COINS is positioned as a diagnostic methodology for synthesized specifications, providing the evaluative foundation needed to interpret the artifacts produced by such autoformalization pipelines.

**Benchmarks for Verifiable Code Generation.** Research has increasingly shifted from evaluating functional correctness toward the formal verification of synthesized code, requiring models to generate implementations that satisfy formal contracts. Dafny-based benchmarks such as Clover (Sun et al., 2024) and DafnyBench (Loughridge et al., 2025) evaluate models on generating verified implementations or auxiliary invariants. In the Lean 4 ecosystem, Clever (Thakur et al., 2025) requires models to jointly generate specifications, implementations, and equivalence proofs against hidden ground truths. VeriBench (Miranda et al., 2026) evaluates end-to-end formal code verification in Lean 4, while its FTP variant focuses on theorem proving for code verification tasks.

More recently, VERINA (Ye et al., 2026) introduces a modular benchmark for code, specification, and proof generation, but its specification evaluation still primarily relies on ground-truth specifications as semantic references. VeriEquivBench (Zeng et al., 2026) replaces ground-truth matching with a Dafny-verified equivalence score over 2,389 algorithmic problems, but still requires proving equivalence between generated code and generated specifications; moreover, its reliance on Dafny's automated SMT-based backend limits the complexity of specifications that can be effectively checked. A further limitation of Lean-based benchmarks such as Clever and VERINA is that their final verified artifacts are Lean implementations and specifications, whereas industrial software is rarely implemented in Lean. In contrast, COINS pairs real-world Python implementations from HumanEval with Rocq specifications, and sidesteps full equivalence proofs by evaluating specifications on instantiated test cases. This test-case-based proving methodology disentangles the Evaluation Duality and provides a more discriminative assessment than binary verification outcomes.

**LLM-based Specification Synthesis.** The automated synthesis of formal properties from code has advanced

from classical dynamic invariant detection (Daikon (Ernst et al., 2007)) to neuro-symbolic and iterative refinement approaches. AutoSpec (Wen et al., 2024) leverages LLMs and static analysis to decompose programs and refine generated specifications. Recent work has further explored LLM-assisted generation of contracts, loop invariants, and assertions: SpecGen (Ma et al., 2025), Clause2Inv (Cao et al., 2025b), NeuroInv (King et al., 2025), SLD-Spec (Chen et al., 2025b), and SESpec (Yang et al., 2025). Beyond generation, other works leverage LLMs to automate test-case construction for validating formal specifications in languages such as Alloy (Cunha & Macedo, 2025) or integrate property-based testing as a core engine for iterative code refinement (He et al., 2025). These works develop sophisticated generation and validation pipelines; COINS contributes a complementary systematic evaluation methodology to analyse the outputs of such systems, enabling an empirical investigation into how specification design choices—such as the Reasoning Gap between operational and relational styles—affect formal verifiability.

**Property-Based Testing and Mutation Analysis for Proof Assistants.** COINS is related to, but distinct from, property-based testing and mutation-analysis tools for proof assistants. The earliest such tool, QuickChick (Paraskevopoulou et al., 2015), tests *executable* properties against generated inputs; in contrast, COINS evaluates potentially non-executable logical specifications by constructing Rocq-checked proof obligations for trusted input-output behaviours. Subsequent Rocq-side mutation-analysis tools, mCoq (Celik et al., 2019) and MutantChick (Cavada et al., 2020), mutate formal artifacts to assess proof-suite adequacy; our mutation step instead mutates Python implementations only, generating negative cases that measure whether a candidate specification is overly permissive.

## 6. Limitations and Future Work

Our current implementation is limited to Rocq specifications for the 164 HumanEval problems. It reflects the substantial cost of manually authoring formal specifications rather than a methodological constraint. The full experimental campaign spanned six frontier models, involving over 100,000 theorem-proving attempts, 620 mutants, and 123,952 positive test cases across the benchmark, where each proof attempt required multi-round LLM compiler interaction with Rocq's type checker. Scaling to other proof assistants or benchmarks demands comparable human annotation and computational effort, which we leave to future work.

Importantly, the COINS methodology is not tied to Rocq or Python, and it applies wherever a test suite and a formal specification language are available. The evaluation pipeline can be readily extended to different implementation languages and proof assistants. In fact, it does not even require a reference implementation: only a natural-language requirement, a function signature, and user-provided positive and negative examples are needed to drive the entire evaluation process. We view COINS as a necessary first step toward our broader goal: fully automated C program verification within the CompCert ecosystem.

## 7. Conclusion

This study suggests that the apparent power of LLMs in program specification synthesis is inseparable from how specification quality is evaluated. The central challenge is therefore not only whether models can generate stronger or more expressive specifications, but whether our evaluation protocols can meaningfully observe and measure that strength. When evaluation is framed solely around whether an implementation satisfies a specification, specification quality itself becomes a latent variable, obscured by the difficulty of formal proof. Strong correctness criteria such as full semantic equivalence further exacerbate this issue by conflating specification quality with prover capability and proof complexity.

Our findings highlight a fundamental asymmetry in formal reasoning: successful proofs constitute reliable positive evidence, while failed proofs are intrinsically ambiguous—they may stem from specification errors, prover limitations, or intractable proof obligations. Ignoring this asymmetry leads to systematic underestimation of model-generated specifications and masks genuine differences in model capability. In contrast, test-case–based formal reasoning provides a pragmatic middle ground, offering sound, interpretable evidence without resorting to brittle, all-or-nothing judgments.

More broadly, this work argues that progress in LLM-based specification synthesis cannot be assessed solely by the ability to discharge equivalence proofs. Instead, future benchmarks should prioritize diagnostic evaluation signals that reflect what contemporary formal tools can establish reliably. By reframing evaluation around provable behavioral evidence, we aim to provide a more faithful answer to how powerful LLMs truly are in generating program specifications, and to enable more principled progress in neural program verification.

## Software and Data

We make our implementation and datasets publicly available at https://github.com/taylor-swift-13/Coins.

## Acknowledgements

This work has been partially funded by the National Natural Science Foundation of China under grant No. 62192732, No. 62432005, W2511064, No. 62572459 and No. 62502475.

## Impact Statement

This paper presents work whose goal is to advance the field of Machine Learning. There are many potential societal consequences of our work, none that we feel must be specifically highlighted here.

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

# A. Negative Test Cases Generation

We supplement positive test cases with automatically generated *negative cases* via mutation testing to detect overly permissive specifications that may incorrectly accept faulty implementations. This Appendix details our negative case generation methodology.

## A.1. Overview

Our approach leverages mutated versions of canonical implementations as oracles for identifying inputs that distinguish correct from incorrect behavior. Given a canonical implementation $f$ and a mutant $f'$, if there exists an input $x$ such that $f(x) \neq f'(x)$, then the pair $(x, f'(x))$ constitutes a *negative case*—a test case on which the original implementation would fail.

## A.2. Mutation Operators

We employ a two-stage mutation strategy: first applying efficient deterministic mutations, then falling back to LLM-guided mutations only when necessary.

**Stage 1: AST-Based Mutations.** Our deterministic mutation engine applies the following operator categories to the Abstract Syntax Tree (AST) of canonical solutions:

- **Binary operators:** $+ \leftrightarrow -, \times \leftrightarrow \div, \% \rightarrow /, // \rightarrow /$
- **Comparison operators:** $== \leftrightarrow != , < \leftrightarrow \geq, > \leftrightarrow \leq$
- **Logical operators:** `and` $\leftrightarrow$ `or`
- **Constants:** $0 \rightarrow 1$, non-zero $\rightarrow 0$, `True` $\leftrightarrow$ `False`

For each task, we generate up to 5 mutants by randomly selecting mutation points. This approach is fast, reproducible, and covers most structural deviations.

**Stage 2: LLM-Guided Mutations.** To complement AST-based mutations, we employ LLMs to generate semantically meaningful mutations that are difficult to express as syntactic rewrites. The LLM is prompted to introduce subtle bugs that preserve syntactic validity while altering program semantics (*e.g.*, off-by-one errors, incorrect boundary handling, or swapped variable references). This stage targets semantic deviations that AST-level operators alone may miss, such as algorithmic logic errors or context-dependent faults.

Together, the two stages are designed to cover both structural and semantic dimensions: Stage 1 provides systematic, reproducible syntactic coverage, while Stage 2 adds harder-to-detect semantic mutants.

## A.3. Negative Case Generation Algorithm

Algorithm 1 presents our negative case generation procedure in two phases.

---

**Algorithm 1** Negative Case Generation via Mutation Testing

---

**Require:** Canonical implementation $f$, test inputs $\mathcal{I}$, mutants $\mathcal{M}$, target $k$
**Ensure:** Negative cases $\mathcal{C}$
1: $\mathcal{C} \leftarrow \emptyset$
2: {Phase 1: Test existing inputs against mutants}
3: **for all** $x \in \mathcal{I}$ **do**
4:     **for all** $f' \in \mathcal{M}$ **do**
5:         **if** $f'(x) \neq f(x)$ **then**
6:             $\mathcal{C} \leftarrow \mathcal{C} \cup \{(x, f'(x))\}$
7:         **end if**
8:     **end for**
9: **end for**
10:
11: {Phase 2: Generate new inputs if needed}
12: **while** $|\mathcal{C}| < k$ **do**
13:     $x' \leftarrow \textsc{Mutate}(\textsc{Random}(\mathcal{I}))$
14:     **for all** $f' \in \mathcal{M}$ **do**
15:         **if** $f'(x') \neq f(x')$ **then**
16:             $\mathcal{C} \leftarrow \mathcal{C} \cup \{(x', f'(x'))\}$
17:         **end if**
18:     **end for**
19: **end while**
20:
21: **return** $\textsc{Sample}(\mathcal{C}, k)$

---

### A.4. Input Variant Generation Strategies

When direct enumeration fails to produce sufficient negative cases, we generate input variants using multiple strategies that cycle every 50 attempts: **(1) Random:** random perturbations; **(2) Small change:** minimal perturbations ($\pm 1$ for integers, $\pm 0.1$ for floats); **(3) Large change:** significant perturbations ($\pm 5$ to $\pm 10$); **(4) Expand:** append elements to collections; **(5) Shrink:** remove elements from collections.

### A.5. Implementation Details

**Execution.** Each code execution is bounded by a 1-second timeout; timed-out executions are treated as errors.

**Input Sources.** We extract inputs primarily from HumanEvalPlus, which provides extended test suites. When unavailable, we fall back to instrumenting the original HumanEval `check` function.

**Deduplication.** Negative cases are deduplicated via hashing of serialized $(input, output)$ pairs to ensure diversity.

### A.6. Statistics

Applying this methodology to the 164 HumanEval tasks yields 1,640 negative cases. All tasks successfully generated the target number.

### A.7. Usage in Evaluation

Negative cases serve as negative examples during specification evaluation. A specification is considered *overly permissive* if it accepts an implementation that produces incorrect outputs on these negative cases.

# B. Theoretical Foundations

We formalize the behavior of a Large Language Model (LLM) acting as a theorem prover within the context of specification verification. This abstraction enables rigorous analysis of the fundamental limitations inherent in using LLMs for formal verification tasks.

## B.1. LLM as a Probabilistic Prover

Consider a specification $S$ defined as a logical predicate over input-output pairs. Given a test case $(i, o)$ consisting of an input $i$ and output $o$, we say that $S$ *accepts* the test case, denoted $S(i, o) = \top$, if the input-output pair satisfies the specification. Conversely, $S$ *rejects* the test case, denoted $S(i, o) = \bot$, if the pair violates the specification.

We model an LLM-based prover as a probabilistic program $\mathcal{M}$ that takes as input a specification $S$ and a test case $(i, o)$, and attempts to produce a proof of acceptance.

**Definition B.1** (LLM Proving Oracle). Let $\mathcal{M}$ be an LLM-based prover. For a specification $S$ and test case $(i, o)$, define the random variable $\text{VERIFY}_{\mathcal{M}}(S, i, o) \in \{\texttt{proof}, \bot\}$ as follows:

- If $S(i, o) = \top$ (true acceptance):

$$\Pr[\text{VERIFY}_{\mathcal{M}}(S, i, o) = \texttt{proof}] = p_{S,i,o} \tag{1}$$
$$\Pr[\text{VERIFY}_{\mathcal{M}}(S, i, o) = \bot] = 1 - p_{S,i,o} \tag{2}$$

  where $p_{S,i,o} \in [0, 1]$ is the *proof success probability*, depending on $S$, $(i, o)$, and $\mathcal{M}$.
- If $S(i, o) = \bot$ (true rejection):

$$\Pr[\text{VERIFY}_{\mathcal{M}}(S, i, o) = \texttt{proof}] = 0 \tag{3}$$
$$\Pr[\text{VERIFY}_{\mathcal{M}}(S, i, o) = \bot] = 1 \tag{4}$$

The second case captures the fundamental *soundness* property: the LLM cannot produce a valid proof for a false proposition, as any generated proof would be rejected by the Rocq type checker.

## B.2. The Undecidability of Verification Failure

A critical consequence of our model is that verification failure is fundamentally uninformative about specification correctness.

**Theorem B.2** (Verification Undecidability). *When* $\text{VERIFY}_{\mathcal{M}}(S, i, o) = \bot$*, it is impossible to determine whether* $S(i, o) = \top$ *or* $S(i, o) = \bot$ *without additional information about* $p_{S,i,o}$.

*Proof.* Consider two scenarios producing identical observable outcomes:

- $S(i, o) = \top$ but $p_{S,i,o}$ is small, i.e., the proof is difficult.
- $S(i, o) = \bot$, i.e., no valid proof exists.

Both yield $\text{VERIFY}_{\mathcal{M}}(S, i, o) = \bot$. Without knowledge of $p_{S,i,o}$, these are observationally indistinguishable. □

This theorem reveals a fundamental Finding 4.1 *evaluation duality*: when verification fails, the fault may lie in either the specification (generator error) or the prover (prover limitation). This provides the formal basis for the "Evaluation Duality" observation in Section 4.1.

A key insight is the *information asymmetry* between outcomes:

- Positive evidence ($\text{VERIFY} = \texttt{proof}$) guarantees $S(i, o) = \top$ by Rocq's soundness.
- Negative evidence ($\text{VERIFY} = \bot$) provides no definitive information.

This asymmetry motivates using successful proof construction as the primary evaluation criterion.

## B.3. Factors Affecting Proof Success Probability

The proof success probability $p_{S,i,o}$ varies based on multiple factors:

- *Proof complexity*: $p_{S,i,o}$ decreases as proof difficulty increases, including proof length, need for auxiliary lemmas, and sophisticated tactics. Our experiments in Section 4.2 support this: all models show substantial drops from $\text{PASS}_{\text{first}}$ to $\text{PASS}_{\text{all}}$ (e.g., Gemini 3 Pro Preview: 60.98% $\to$ 28.05%).

- *Specification style*: executable components (`Fixpoint`) often admit simpler proofs via computation, increasing $p_{S,i,o}$; pure relational specifications (`Inductive`) may require explicit inductive reasoning, decreasing $p_{S,i,o}$. This is empirically validated in Section 4.4.

- *Test case complexity*: larger inputs or edge cases require more intricate proof steps, reducing $p_{S,i,o}$.

- *Model capability*: different LLMs exhibit varying $p_{S,i,o}$ for the same task, as shown in Table 1.

## B.4. Formal Analysis of the COINS Framework

We provide formal justification for our evaluation stages (Section 3). To distinguish *metrics* (percentages in experiments) from *predicates* (boolean functions), we use:

- $\mathcal{P}_{\text{all}}(S), \mathcal{R}_{\text{all}}(S) \in \{\top, \bot\}$: predicates on specification $S$.
- $\text{PASS}_{\text{all}}, \text{REJECT}_{\text{all}}$: metrics representing the proportion of specifications satisfying the predicates.

**The $\mathcal{P}_{\text{all}}$ predicate.** Let $\mathcal{T}^+ = \{(i_1, o_1), \ldots, (i_n, o_n)\}$ be the positive test cases. Define:

$$\mathcal{P}_{\text{all}}(S) = \top \iff \forall (i, o) \in \mathcal{T}^+ : \text{VERIFY}_{\mathcal{M}}(S, i, o) = \texttt{proof} \tag{5}$$

The metric $\text{PASS}_{\text{all}} = |\{S : \mathcal{P}_{\text{all}}(S) = \top\}|/N$ where $N$ is the total number of specifications. By Rocq's soundness, $\mathcal{P}_{\text{all}}(S) = \top$ implies $S$ accepts all positive test cases: $\forall (i, o) \in \mathcal{T}^+ : S(i, o) = \top$. Note that $\mathcal{P}_{\text{all}}(S) = \bot$ does not imply incorrectness—it may indicate low $p_{S,i,o}$ for some test cases.

**The $\mathcal{R}_{\text{all}}$ predicate.** Let $\mathcal{T}^- = \{(i_1', o_1'), \ldots, (i_m', o_m')\}$ be the negative cases (Section 3.1). Define:

$$\mathcal{R}_{\text{all}}(S) = \top \iff \forall (i', o') \in \mathcal{T}^- : \text{VERIFY}_{\mathcal{M}}(S, i', o') = \bot \tag{6}$$

If $\text{VERIFY}_{\mathcal{M}}(S, i', o') = \texttt{proof}$ for any $(i', o') \in \mathcal{T}^-$, then $S$ is provably overly permissive. Conversely, $\mathcal{R}_{\text{all}}(S) = \top$ does not guarantee $S$ rejects all negative cases—the LLM may fail to find proofs even when they exist.

**Candidate specifications.** A specification $S$ with $\mathcal{P}_{\text{all}}(S) = \mathcal{R}_{\text{all}}(S) = \top$ is a *candidate specification* (corresponding to specifications passing $\text{REJECT}_{\text{all}}$ in Table 2). Such specifications:

- provably accept all positive test cases;
- are not provably overly permissive;
- may still be incorrect in ways not captured by the test suite.

## B.5. Why Full Equivalence Proving is Impractical

Consider proving equivalence $\forall i, o : S_1(i, o) \Leftrightarrow S_2(i, o)$. Unlike test-case verification, this requires handling arbitrary inputs through induction. Let $p_{\text{equiv}}$ denote the success probability; empirically $p_{\text{equiv}} \ll p_{S,i,o}$.

Our results in Section 4.3 confirm this: among specifications passing all test-based criteria, only 31.0% yield provably isomorphic specifications. This illustrates the Finding 4.3 *verification complexity trade-off*: as verification complexity increases, proof success probability decreases, causing systematic underestimation of specification quality.

## B.6. Design Principles

Our analysis yields four design principles:

**Exploit asymmetry.** Maximize use of positive evidence while acknowledging uncertainty in negative evidence.

**Calibrate difficulty.** Avoid verification tasks that are too difficult; overly complex proof obligations lead to uniformly low $p_{S,i,o}$, preventing meaningful differentiation between specifications.

**Use strongest prover.** Maximize $p_{S,i,o}$ to minimize false negatives from prover limitations.

**Anchor the duality.** Use known-good artifacts (human specifications) to anchor assessment.

The COINS framework embodies these principles through test-case-based verification, Gemini 3 Pro Preview as prover (Section 4.1), human specification anchors (Section 4.3), and epistemic humility about failed proofs.

## C. Experimental Configuration Details

### C.1. Model Configuration

Table 3 summarizes the model versions and API identifiers used in our experiments, together with their default reasoning configurations as provided by the respective platforms. Unless otherwise specified, we use the default inference settings recommended by each provider.

For specification generation, we set the temperature to 0 to ensure deterministic and reproducible outputs. For proof generation, we set the temperature to 0.7 to balance precision with exploratory diversity, as the task involves multiple rounds of Rocq feedback and revision.

### C.2. Prompt Design

Our prompt design follows a deliberately minimalistic principle. Rather than providing detailed heuristics, step-by-step guidance, or domain-specific proof strategies, we restrict prompts to essential task descriptions and formatting constraints. This design avoids injecting redundant or overly prescriptive instructions, allowing models to independently determine what constitutes a suitable specification and a valid proof.

By minimizing prompt-level guidance, we aim to evaluate the models' intrinsic ability to synthesize high-quality specifications and construct correct proofs, rather than their capacity to follow handcrafted prompting strategies. All prompts used in the experiments are reported below:

*Table 3.* Model configurations used in our experiments.

| Model | API Identifier | Default Reasoning |
|---|---|---|
| Gemini 3 Pro Preview | `gemini-3-pro-preview` | High |
| Claude 3.7 Sonnet | `claude-3-7-sonnet-20250219` | Non-reasoning |
| Claude 4.5 Opus | `claude-opus-4-5-20251011` | Non-reasoning |
| GPT-4o | `gpt-4o` | – |
| GPT-5 | `gpt-5` | Medium |
| DeepSeek V3.1 | `deepseek-v3.1` | Non-reasoning |

---

**Specification Generation Prompt**

```
You are an expert in writing Rocq specifications.
Below is a textual description of a program and its corresponding Python code.
Your task is to generate a precise and complete Rocq specification
for this program based on the provided information.
Rules
- Including the Require Import statements.
- Do not generate any natural language comments.

Before generating the output, please refer to the following example:
[Example]
    [Input]
        Now, this is the description:

        def add(A: int, B: int):
            """
            return the sum of A and B.
            """
        ,
        and this is the code:

            return A + B
        .
    [Output]
        ```coq
        Definition add_spec (A : int) (B : int) (sum : int) : Prop :=
        sum = A + B.
        ```

Now, this is the description: {input_text},
and this is the code: {code}.
Please output only the Rocq specification for this program.
```

**PASS_first Prompt**

```
Please generate a complete Rocq proof for the following specification and test
case.
Specification definition:
{spec_content}
Test case:
{example}
Please generate an Example proof, including the full Proof steps and ending
with Qed. If additional imports or lemmas are required, please also provide
them.

Notes:
1. Use appropriate Rocq tactics (such as unfold, simpl, reflexivity etc.)
2. Ensure the proof is complete and correct
3. If real number operations are involved, please use R_scope
4. If integer operations are involved, please use Z_scope
5. For list operations, please use the appropriate List library functions

Please return Rocq code directly, without any additional explanations.
```

**PASS_all Prompt**

```
You are given the full content of an existing Rocq output file for a HumanEval
spec. It already includes the specification definitions and the proof for the
first test case.

Existing Rocq output file content
specification for the first test case '{first_test_case}':
```coq
{existing_output_content}
```
Now, generate a Rocq Example proof for the following new test case {test_case}
and replace the first test case:

Requirements:

Return the full content of the Rocq output file for the new test case.

Do NOT add any natural language comments.

End with Qed.
```

---

**REJECTEDall Prompt**

```
Please generate a complete Rocq proof for the following specification and test
case.

Specification definition:
{spec_content}

Test case:
{example}

First, judge whether you can generate a valid proof for the given spec and
test case.
- If YES: Generate the complete Example proof (Proof steps + Qed), with any
necessary imports or lemmas.
- If NO: Return only "Test case is not accepted".

Do NOT modify the given spec or test case.

Notes:
1. Use appropriate Rocq tactics (such as unfold, simpl, reflexivity etc.)
2. Ensure the proof is complete and correct
3. If real number operations are involved, please use R_scope
4. If integer operations are involved, please use Z_scope
5. For list operations, please use the appropriate List library functions

Please return Rocq code directly, without any additional explanations.
```

---

**Equivalence Prompt**

```
You are a Rocq proof assistant. Given two Rocq specifications, your task is to
generate a COMPLETE, STANDALONE Rocq file that proves the first specification
logically implies the second specification.

First specification ({spec1_name}):
```coq
{spec1}
```
Second specification ({spec2_name}):
```coq
{spec2}
```
Your task is to generate a COMPLETE Rocq file that:
Includes ALL necessary Require Import statements from both specifications
Includes ALL definitions from the first specification
Includes ALL definitions from the second specification
States a theorem that expresses the implication relationship
Provides a complete proof using appropriate Rocq tactics
Is syntactically correct and can be verified by Rocq independently
Rules:

The generated code must be a COMPLETE, STANDALONE Rocq file

Generate ONLY Rocq code, without any explanations
```

**Error Feedback Prompt**

```
IMPORTANT: The previous proof attempt failed with the following error:
{error_info}

Please fix the proof based on this error information and generate a corrected
version.
```

## D. Statistics of LLM-Generated Specifications

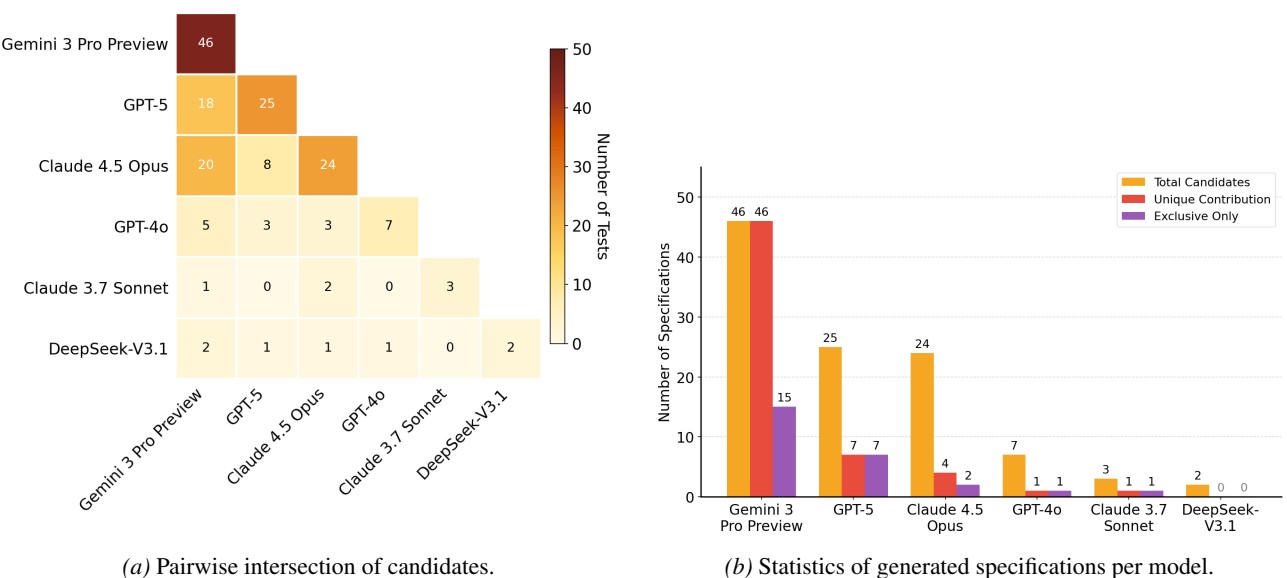

*(a)* Pairwise intersection of candidates.    *(b)* Statistics of generated specifications per model.

*Figure 5.* Model performance on candidate specification generation. (a) Diagonal entries show each model's total candidates; off-diagonal entries show shared successes. (b) Total candidates, unique contributions to the aggregated set, and exclusive successes (solved by that model only).

Figure 5 reports detailed statistics of candidate specifications generated by each LLM. We analyze both the total number of valid specifications per model and their overlap structure to better understand model complementarity under our aggregation strategy.

Figure 5a presents the pairwise intersection matrix. Diagonal entries correspond to the total number of valid specifications produced by each model, while off-diagonal entries indicate shared successes. The results reveal substantial but incomplete overlap among top-performing models. Gemini 3 Pro Preview shares 18 specifications with GPT-5 and 20 with Claude 4.5 Opus, reflecting its broad coverage. In contrast, GPT-5 and Claude 4.5 Opus share only 8 specifications despite having comparable total counts, indicating that they capture different regions of the specification space. Lower-performing models exhibit limited overlap both with each other and with the top-tier models, and their successes are largely subsumed by stronger models.

Figure 5b further decomposes each model's contribution into total candidates, unique contributions to the aggregated set, and exclusive successes (i.e., problems solved only by that model). Gemini 3 Pro Preview achieves the strongest performance with 46 valid specifications, including 15 exclusive successes not covered by any other model. GPT-5 and Claude 4.5 Opus generate 25 and 24 valid specifications, respectively; however, GPT-5 contributes more exclusive specifications (7 vs. 2), suggesting greater diversity relative to other models. The remaining models—GPT-4o, Claude 3.7 Sonnet, and DeepSeek-V3.1—produce only a small number of valid specifications and add limited unique coverage.

Overall, these statistics motivate our priority-based aggregation strategy described in the main text. Although individual models differ in overall capability, their partial overlap and complementary strengths imply that no single model dominates across all problems. Aggregating candidates by model ranking therefore improves overall coverage while retaining the

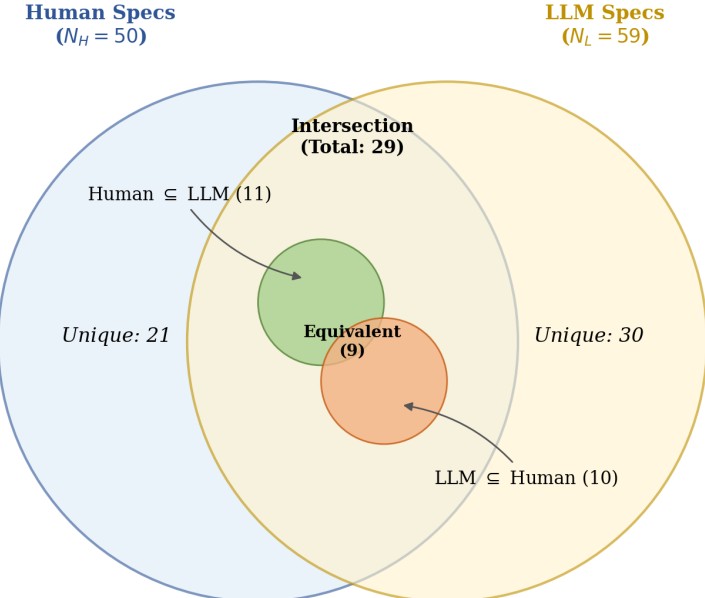

*Figure 6.* Equivalence analysis between human ($N_H = 50$) and LLM ($N_L = 59$) specifications. Among 29 overlapping problems, only 9 yield provably equivalent specifications.

highest-quality specification available for each problem.

## E. RQ5: How Do LLM-Generated Specifications Compare to Human-Written Ones?

To assess the quality of LLM-generated specifications relative to human-written ones, we conduct a systematic comparison using equivalence proofs. We first aggregate candidate specifications from all models, then attempt to prove logical equivalence between LLM-generated and human-written specifications.

**Candidate LLM-Generated Specification Aggregation.** To maximize coverage, we aggregate candidates using priority-based selection: for each problem, we select the specification from the highest-ranked model (by REJECT_all). This yields 60 unique specifications, covering 36.6% of the benchmark. Detailed statistics are reported in Appendix D.

**Candidate Human Specifications.** We validate human-written specifications through Gemini 3 Pro Preview's REJECT_all criterion, yielding 50 candidates (30.5% coverage)—comparable to the 59 LLM-generated specifications (36.0%), suggesting frontier models can produce formal specifications at a rate approaching human experts.

**Equivalence Analysis.** Figure 6 shows the relationship between 50 validated human and 59 LLM specifications. Of 80 unique problems covered, only 29 (37.2%) have valid specifications from both sources. Within this intersection, 9 (31.0%) yield provably equivalent specifications with both implication directions formally verified, 3 cases succeed in only one direction, and the remaining 17 cases fail to prove either direction. We manually inspected all 3 single-direction cases and confirmed that the specifications are indeed semantically equivalent—the missing direction fails due to prover limitations rather than genuine semantic differences. This confirms Finding 4.3 that equivalence proving introduces systematic underestimation and is unsuitable as a primary quality metric.

The limited overlap between human and LLM successes suggests complementary strengths: they succeed on distinct problem subsets rather than merely differing in capability level, indicating LLM-based specification synthesis remains a challenging open problem.

## F. RQ6: Is Formal Specification Generation More Like Math or Code?

Figure 7 reveals a striking divergence between established benchmark performance and formal specification quality. While newer frontier models consistently outperform their predecessors on LiveCodeBench, SWE-bench Verified, and AIME 2025—with Gemini 3 Pro Preview (91.7%, 74.2%, 95.7%) and GPT-5 (70.3%, 65.0%, 91.7%) leading across the board, and

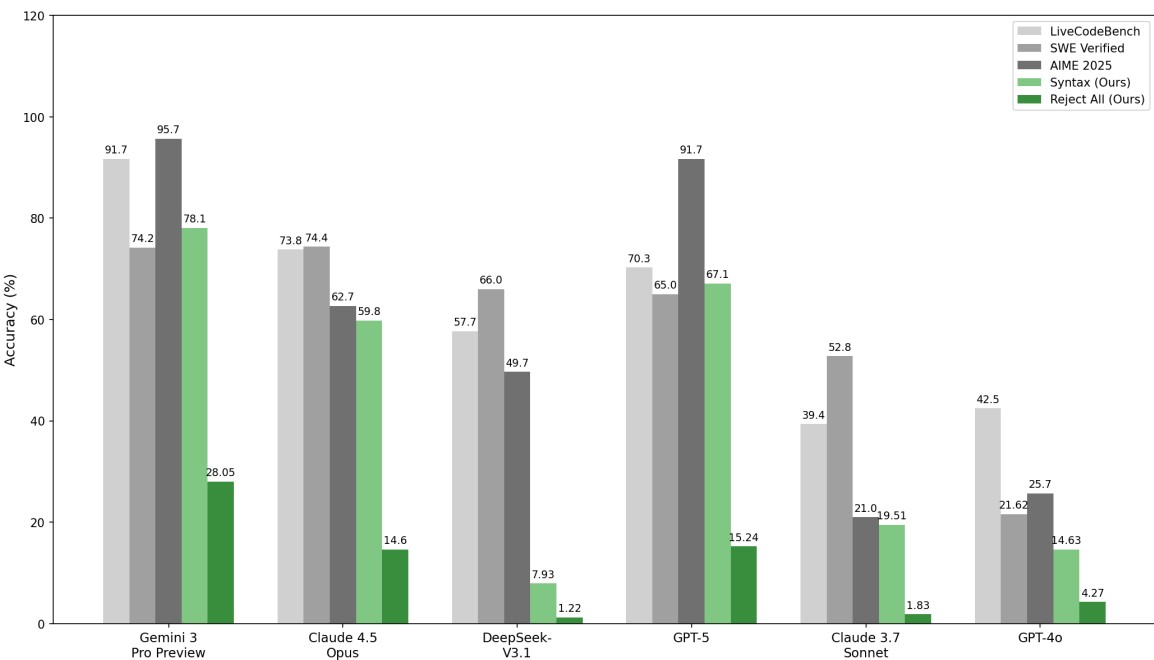

*Figure 7.* Comparison of model performance across code generation (LiveCodeBench, SWE-bench Verified), mathematical reasoning (AIME 2025), and formal specification generation (Syntax and Reject All on our benchmark). Models are ordered by release date (newest first).

older models like GPT-4o (42.5%, 21.62%, 25.7%) and Claude 3.7 Sonnet (39.4%, 52.8%, 21.0%) trailing behind—this generational improvement does not reliably transfer to our specification metrics.

Most notably, DeepSeek-V3.1 exemplifies this disconnect: despite strong performance on SWE-bench Verified (66.0%)—comparable to GPT-5 (65.0%) and surpassing Claude 3.7 Sonnet (52.8%)—it achieves the lowest Syntax score (7.93%) and Reject All score (1.22%) among all evaluated models. Similarly, Claude 4.5 Opus achieves the highest SWE-bench score (74.4%) yet underperforms on both Syntax (59.8%) and Reject All (14.6%) compared to Gemini 3 Pro Preview (78.1%, 26.2%). Interestingly, Gemini 3 Pro Preview and GPT-5, the two models with the strongest AIME 2025 scores (95.7% and 91.7%), also achieve the best specification quality, suggesting that formal specification generation may be more closely related to mathematical reasoning than to software engineering and coding capabilities.

Remarkably, even on HumanEval—a benchmark where frontier models have long achieved near-saturation on code generation—our specification metrics reveal substantial performance gaps among top-tier models: 18.3% on Syntax and 11.6% on Reject All, comparable to the differentiation observed on challenging benchmarks like LiveCodeBench (21.4% gap) and SWE-bench Verified (9.4% gap). This demonstrates that generating tight formal specifications remains a discriminative task that exposes meaningful capability differences, even when the underlying programming problems are considered solved.

## G. RQ7: What Are the Principal Barriers to Syntactic Correctness in Rocq?

While state-of-the-art models achieve relatively high syntax compliance, a non-negligible fraction of generated specifications still fails to compile (as reflected in the SYNTAX metric in Table 2). To understand the root causes, we conducted a qualitative analysis of all 156 specifications that failed the SYNTAX stage across the top three performing models: Claude 4.5 Opus, GPT-5, and Gemini 3 Pro Preview. We classified the errors into three primary categories:

- **Type Mismatches (45%):** Logical errors where the model conflates distinct Rocq types, such as Peano natural numbers (`nat`), binary integers (`Z`), and propositions (`Prop`).
- **Missing Dependencies (22%):** Cases where the model invokes semantically appropriate functions (e.g., from `Ascii` or `ZArith`) but fails to include necessary `Require Import` statements.
- **Ill-formed Recursion (17%):** `Fixpoint` definitions that violate the *Calculus of Inductive Constructions*' strict

structural termination requirements, often by recursing on non-syntactic sub-terms.

**Error Distribution and Analysis.** Table 4 summarizes the error distribution. These failures reflect a fundamental tension between the probabilistic nature of LLMs and the rigid logical kernels of proof assistants.

*Table 4.* Breakdown of compilation errors (N=156) by category.

| Model | Type Mismatch | Missing Dep. | Bad Recursion | Others | Total |
|---|---|---|---|---|---|
| Claude 4.5 Opus | 24 | 19 | 12 | 11 | 66 |
| GPT-5 | 29 | 7 | 9 | 9 | 54 |
| Gemini 3 Pro Preview | 17 | 8 | 5 | 6 | **36** |
| **Total** | **70** (45%) | **34** (22%) | **26** (17%) | **26** (17%) | 156 |

**Type System Rigidity vs. Language Bias.** Type Mismatches remain the dominant failure. Models frequently treat `nat` and `Z` interchangeably, likely an artifact of pre-training on dynamically typed languages like Python. Unlike those languages, Rocq requires explicit casting (e.g., `Z.of_nat`) due to the fundamentally different inductive structures of these types.

**Context Management and Imports.** Claude 4.5 Opus exhibited a notably higher rate of Missing Dependencies (19 cases). This suggests a "knowledge-action gap": the model understands the semantic need for a library function but neglects the syntactic boilerplate required to make it available in the current environment. Gemini 3 Pro Preview demonstrated more robust context awareness in this regard.

**The Structural Termination Gap.** All models struggle with Ill-formed Recursive Definitions (17% of errors). Rocq's guard checker requires recursive functions to be structurally terminating, meaning recursive calls must be performed on direct syntactic sub-terms of the inductive input. We observed that models often generate "Pythonic" recursion patterns that are mathematically sound but formally rejected.

In this section, we provide a concrete example of the **Ill-formed Recursion** failure mode, which accounts for approximately 17% of compilation errors. Figure 8 illustrates a case where the model attempts to generate a specification for a sequence similar to Fibonacci (specifically, the `fib4` sequence).

## H. RQ8: Does Input Data Complexity Dictate the Success of Formal Specification Generation?

To investigate whether the complexity of the input data structure influences the difficulty of generating precise specifications, we categorized the 164 HumanEval problems based on their input argument types. We analyze the relationship between these input categories and the models' ability to generate **candidate specifications**—defined as specifications that satisfy the REJECT$_{all}$ criterion (i.e., they accept all positive test cases and successfully reject negative test cases).

We grouped the input types into five distinct categories:

- **Basic Numeric:** Scalars involving `Z`, `nat`, `float`, and `R`.
- **String:** Pure string manipulation tasks.
- **List Numeric:** Lists containing numeric types (`list Z`, `list nat`, etc.).
- **List String:** Lists of strings (`list string`).
- **Complex/Nested:** Nested structures (e.g., `list list Z`), dictionaries, or custom types.

We categorize the problems into three groups based on model performance:

- **Weak Accessible**: Problems where at least one of the "Weak Models" (GPT-4o, Claude 3.7 Sonnet, DeepSeek-V3.1) successfully generated a candidate specification;
- **Strong Exclusive**: Problems where valid candidates were produced *only* by the "Strong Models" (Gemini 3 Pro Preview, GPT-5, Claude 4.5 Opus);
- **No Valid Spec**: Problems where no model across either group achieved the REJECT$_{all}$ criterion.

**Case Study: Ill-formed Recursion in `fib4_seq`**

**Generated Specification:**

```
Require Import Coq.ZArith.ZArith.
Require Import Coq.Arith.PeanoNat.

Open Scope Z_scope.

Fixpoint fib4_seq (n : nat) : Z :=
  match n with
  | 0%nat => 0
  | 1%nat => 0
  | 2%nat => 2
  | 3%nat => 0
  | S (S (S (S m))) => fib4_seq (S (S (S m))) + fib4_seq (S (S m)) +
                       fib4_seq (S m) + fib4_seq m
  end.

Definition fib4_spec (n : Z) (res : Z) : Prop :=
  n >= 0 /\ res = fib4_seq (Z.to_nat n).
```

**Rocq Compiler Error:**

```
Compiler Output

Recursive definition of fib4_seq is ill-formed.
In environment
fib4_seq :  nat -> Z
...
Recursive call to fib4_seq has principal argument equal to "S m" instead of one of
the following variables:  "n0" "n1" "n2" "m".
```

*Figure 8.* **Example of Ill-formed Recursion.** The model attempts to implement a recurrence relation $f(n) = \sum_{i=1}^{4} f(n-i)$ but fails Rocq's structural termination check. Rocq requires recursive arguments to be strict subterms of the matched pattern (e.g., m), but the model passes S m (conceptually $n-3$), which Rocq cannot automatically verify as terminating without additional measures.

## H.1. Quantitative Results

Table 5 presents the distribution of valid specifications across input types. The data reveals a strong correlation between input complexity and the ability to generate sufficiently precise specifications that pass the REJECT$_{all}$ filter.

## H.2. Analysis of Failure Modes

**The String Gap: Frontier Models' Moat.** The most striking disparity appears in the **String** category. While weak models achieved a negligible 4.1% success rate in generating valid candidates, strong models exclusively covered 40.8% of string problems. In Rocq, specifying string properties requires handling the inductive `string` type (essentially a list of ASCII characters). Weak models often fail to satisfy REJECT$_{all}$ because they generate overly permissive specifications (e.g., using Pythonic logic that doesn't strictly constrain the Rocq string behavior) or fail to construct the necessary inductive predicates to rule out incorrect implementations. Frontier models demonstrate a superior ability to formalize these structural constraints precisely.

**Arithmetic vs. Structural Reasoning.** Weak models perform best in the **Basic Numeric** category (6 out of their 11 successes). This suggests that their ability to generate precise specifications is largely limited to First-Order Arithmetic. However, performance drops precipitously when shifting to **List Numeric** types (3.6% success). For lists, passing REJECT$_{all}$ requires the specification to be strict enough to reject incorrect list manipulations (e.g., off-by-one errors or incorrect

*Table 5.* Distribution of Valid Candidate Specifications (Passing REJECT$_{all}$) by Input Type. **Weak Accessible** denotes problems where at least one weak model produced a valid spec. **Strong Exclusive** denotes problems where only strong models produced valid specs. **No Valid Spec** denotes problems where no model passed REJECT$_{all}$.

| INPUT CATEGORY | TOTAL | WEAK ACCESSIBLE | STRONG EXCLUSIVE | NO VALID SPEC |
|---|---|---|---|---|
| BASIC NUMERIC | 43 | 6 (14.0%) | 9 (20.9%) | 28 (65.1%) |
| STRING | 49 | 2 (4.1%) | **20 (40.8%)** | 27 (55.1%) |
| LIST NUMERIC | 56 | 2 (3.6%) | 16 (28.6%) | 38 (67.9%) |
| LIST STRING | 9 | 1 (11.1%) | 1 (11.1%) | 7 (77.8%) |
| COMPLEX/NESTED | 7 | 0 (0.0%) | 2 (28.6%) | 5 (71.4%) |
| **TOTAL** | **164** | **11 (6.7%)** | **48 (29.3%)** | **105 (64.0%)** |

permutations). Weak models struggle to synthesize the complex 'Forall' or recursive predicates needed to tightly constrain list behaviors, resulting in specifications that are either too loose (failing REJECT$_{all}$) or syntactically invalid.

**The Complexity Barrier.** As input complexity increases to **List String** and **Complex/Nested** types, the failure rate for generating valid specifications rises to over 70% across all models. Problems involving nested data structures (e.g., `list list nat`) require specifications that handle multiple layers of induction. The high failure rate in the "No Valid Spec" column for these categories indicates that even the strongest models currently struggle to generate formal specifications that are precise enough to distinguish correct implementations from incorrect ones when the data structure complexity exceeds basic scalar or linear types.

# I. RQ9: Why Do LLMs Fail to Prove Valid Specifications on Test Cases?

While Gemini 3 Pro Preview demonstrates impressive capability in verifying individual test cases (achieving 74.39% on PASS$_{first}$ with human specifications), its performance drops significantly when scaling to the full test suite (PASS$_{all}$). This gap suggests that verifying a diverse set of inputs introduces complexities that are not present in the first (often simple) test case.

Through a qualitative analysis of failed verification attempts where the specification was known to be correct (or passed the first case), we identified three recurrent failure modes specific to the LLM's behavior as a theorem prover.

## I.1. Phantom Goal Errors: Tactic-State Mismatch

A frequent error occurs when the model attempts to discharge verification conditions involving multiple logical conjunctions (e.g., $A \wedge B \wedge C$). The model often employs the `repeat split` tactic, assuming it must explicitly prove every sub-goal. However, Rocq's automation (specifically `split` combined with internal simplifications) often automatically discharges trivial sub-goals immediately.

As illustrated in Figure 9, the model anticipates three sub-goals corresponding to the specification structure. It generates a proof script with three bullets. However, Rocq solves the first two sub-goals instantly, leaving only the third complex goal. The model then attempts to apply the tactic for the first phantom goal (e.g., `reflexivity`) to the actual remaining goal (e.g., a `forall` quantifier), causing a unification error.

## I.2. The Library Barrier: Complex Inductive Predicates

When specifications involve complex standard library predicates—specifically `Permutation` (for sorting/reordering tasks) or `NoDup` (for uniqueness constraints)—the model's proof capability degrades sharply. Unlike basic arithmetic which can often be solved by `lia`, proving properties about permutations requires selecting the correct lemmas from the `Coq.Sorting.Permutation` library. The model frequently attempts to unfold these definitions and prove them from scratch using low-level tactics, which leads to unmanageably long proof contexts or getting stuck in inductive steps that require auxiliary lemmas.

As illustrated in Figure 10, the model often fails to utilize high-level automation and instead resorts to brittle manual proof constructions.

---

**Failure Mode 1: The Phantom Goal Error (HumanEval/10)**

**Specification Snippet:**

```
Definition problem_10_spec (input output : string) : Prop :=
  prefix input output = true /\        (* Part 1: Trivial for empty string *)
  palindrome output /\                 (* Part 2: Trivial for empty string *)
  forall p : string, ...               (* Part 3: Complex minimality condition *)
```

**The Mismatch:**

```
(* Test case: input = "", output = "" *)
Example test_problem_10_empty : problem_10_spec "" "".
Proof.
  unfold problem_10_spec.
  repeat split.
  (* AT THIS POINT: Rocq solves Part 1 & 2 automatically. *)
  (* Only Part 3 remains. But the model thinks all 3 goals exist. *)

  - (* Model attempts to prove Part 1 (prefix "" "" = true) *)
    simpl.
    reflexivity.
    (* ERROR: Rocq tries to apply reflexivity to Part 3 (forall p...) *)

  - (* Model code for Part 2... (Dead code) *)
  - (* Model code for Part 3... (Dead code) *)
Qed.
```

**Rocq Compiler Error:**

**Compiler Output**

```
In environment...
Unable to unify "String.length p" with "0".
```

---

*Figure 9.* **Tactic-State Mismatch Example.** The model hallucinates a proof state with three sub-goals. Because Rocq automatically solved the first two, the model applies the tactic for the first goal (`reflexivity`) to the third goal (an inequality), leading to a unification failure.

### I.3. Resource Exhaustion on Large Inputs

The PASS_all phase introduces generated test cases with significantly larger numerical values than the canonical examples. When specifications are defined using Peano natural numbers (`nat`) or structurally recursive functions without tail-call optimization, verifying large inputs triggers **Stack Overflow** errors during Rocq's normalization process (e.g., using `simpl` or `compute`). Even when using binary integers (`Z`), models sometimes generate inefficient proof scripts that attempt to unfold computations explicitly rather than using efficient reduction strategies like `vm_compute`, causing the verifier to crash or time out.

## J. RQ10: Does Syntactic Validity Guarantee Semantic Correctness?

A core motivation for our execution-based evaluation is the observation that syntactic validity (compilation) is a necessary but insufficient condition for specification quality. In our experiments, we encountered cases where LLMs generated specifications that successfully compiled in Rocq but contained fundamental semantic flaws. These flaws often manifest as logical contradictions or overly restrictive constraints that rule out valid program behaviors.

This phenomenon highlights a critical failure mode: the model knows how to write valid Rocq code structure but fails to translate the natural language intent into a consistent logical reality. Because these specifications are syntactically perfect, they pass initial static checks, making them particularly deceptive. They only reveal their defects when instantiated against concrete test cases during the verification phase.

---

**Failure Mode 2: Inability to Discharge Library Predicates (HumanEval/33)**

**Specification Snippet:**

```
Definition problem_33_spec (input output : list Z) : Prop :=
  (* 1. input is a Permutation of output *)
  Permutation input output /\
  (* 2. values at non-divisible indices are preserved *)
  (forall (i : nat), ...) /\
  (* 3. elements at indices divisible by 3 are sorted *)
  (forall (i j : nat), ...).
```

**The Mismatch:**

```
Example test_case_problem_33 : problem_33_spec [2; 10; 20...] [2; 10; 20...].
Proof.
  unfold problem_33_spec.
  split.
  - (* Part 1: Permutation. Model tries manual construction *)
    apply Permutation_cons.
    (* This tactic creates subgoals: head equality (2=2) and tail Permutation *)
    apply Permutation_cons.
    (* ERROR: Model applies Permutation_cons again, but likely
       focused on the equality subgoal (2%Z = 2%Z) instead of the list *)
    apply Permutation_trans.

  - (* Part 2... *)
```

**Rocq Compiler Error:**

**Compiler Output**

```
Unable to unify
"forall x y :  ?M1435, x = y -> Morphisms.respectful ..."
with "2%Z = 2%Z".
```

---

*Figure 10.* **Library Predicate Failure.** Instead of using automation (e.g., `Permutation_refl`), the model attempts a brittle manual proof. The error occurs when it tries to apply a Permutation lemma to a subgoal that requires proving numerical equality ($2\%Z = 2\%Z$), revealing a lack of awareness regarding the proof state evolution of complex library predicates.

Figure 11 presents a representative case study from HumanEval/12 (`longest`). The task requires returning the first string of maximum length. The specification generated by Claude 4.5 Opus compiles without errors. However, it introduces a subtle logic error in how it defines the first occurrence. The specification inadvertently enforces a constraint that requires the result's index to be smaller than or equal to *any* index of a max-length string. As shown in the trace, this logic collapses when the input list contains duplicate max-length strings (e.g., `["a", "a"]`), creating a mathematical contradiction ($1 \leq 0$). This case underscores that specification generation is not just a translation task but requires rigorous semantic consistency, which LLMs still struggle to guarantee.

**Case Study: Syntactically Valid but Semantically Incorrect Spec**

**Generated Specification (Compiles Successfully):**

```
Definition longest_spec (strings : list string) (result : option string) : Prop :=
  match strings with
  | [] => result = None
  | _ => let maxlen := max_length strings in
    exists s, result = Some s /\ In s strings /\ string_length s = maxlen /\
      (* Logic Error: Universal quantifier creates contradiction for duplicates *)
      (forall s'' idx1 idx2,
        In s'' strings -> string_length s'' = maxlen ->
        nth_error strings idx1 = Some s ->   (* s is the result *)
        nth_error strings idx2 = Some s'' -> (* s'' is any max-len string *)
        idx1 <= idx2)
  end.
```

**Why the Logic Fails (Trace):**

```
(* Input: strings = ["a"; "a"], maxlen = 1. Implementation returns index 0 ("a"). *)
(* Spec requires condition for ALL index pairs pointing to max-len strings. *)
(* Note: "a" appears at both index 0 and index 1. *)

Variable idx1 := 1. (* "a" at index 1 *)
Variable idx2 := 0. (* "a" at index 0 *)

(* The Spec Constraint: idx1 <= idx2 *)
(* Substitution:        1 <= 0        *)
(* Result: False. Verification fails regardless of prover power. *)
```

*Figure 11.* **Example of Semantic Error in HumanEval/12.** The model generates a Rocq specification that is syntactically correct but semantically flawed. It attempts to formalize the first longest string requirement but introduces a logic error that forbids duplicates. This results in a specification that is logically unsatisfiable for certain valid inputs, illustrating the gap between syntax generation and semantic understanding.

