# OpenReview forum: "How Powerful are LLMs in Generating Formal Program Specifications?"
_ICML.cc/2026/Conference — ICML 2026 regular_

### Official Review · Reviewer_EP9M · 2026-02-16

**Soundness:** 3
**Presentation:** 3
**Significance:** 2
**Originality:** 3
**Overall Recommendation:** 4
**Confidence:** 4

**Summary:**

The paper introduces COINS, a Coq-based evaluation framework to assess how powerful LLMs are in generating Coq specifications for Python programs. The framework is accompanied by a curated dataset of human-written Coq specifications for 164 HumanEval problems and proceeds in several steps. First, the LLM is prompted to generate a formal specification for one of the 164 programs. Then, four checks are performed in successive order to minimize the computational effort: (1) checking that a generated specification is syntactically well-formed and successfully compiles to Coq; (2) checking the valid specification for consistency with a single positive test case; (3) checking the specification for consistency with the remaining positive test cases; and (4) evaluating the specification against automatically generated negative test cases. COINS is applied to all 164 benchmark programs, revealing that many state-of-the-art LLMs struggle to produce correct formal program specifications.

**Compliance With Llm Reviewing Policy:**

Affirmed.

**Ethical Review Concerns:**

The paper addresses challenges in software verification. I do not see any ethical concerns.

**Final Justification:**

While the generation of specifications is a relevant research problem and the paper proposes an original approach, the work is limited in scope (addressing only Python and Coq), which reduces its potential significance. Nonetheless, I believe the paper provides a solid foundation for further research in this area. The presentation and technical soundness are good.

The rebuttal clarified that a mere translation of a Python program into Coq does not solve the overall problem. As a consequence, I have raised my score. Nonetheless, I still believe that the submission is among the weaker papers. Expanding the paper's scope and re-submitting it to the next ML conference would substantially improve its impact.

**Key Questions For Authors:**

1. Is the program itself a proper specification? If so, how do you ensure that you do not measure how well an LLM can translate a Python program into Coq?

**Limitations:**

The paper does not seem to have negative societal impact.

The paper seems more limited than the title suggests (they focus on formal specification in Coq for Python programs). The authors never hide this fact, but should make the limitation clearer. Further comments can be found in the sections on strength and weaknesses.

**Strengths And Weaknesses:**

# Strengths
- The paper addresses an interesting problem and provides a promising (albeit somewhat limited solution).
- The paper builds a good foundation for further research in the area.

# Weaknesses
- The paper only investigates specification in the input language for the Coq interactive proof assistant and focuses exclusively on Python programs; other verification settings are left out.
- It is not clear whether the proposed framework measures how well LLMs can generate formal specifications in Coq or how well they can translate programs into Coq.
Review

# Detailed comments
Generating formal specifications is a central challenge in computer-aided verification and remains, to this date, a manual, error-prone task. Therefore, the use of LLM to automate this task is highly promising, yet a systematic evaluation of LLM's capabilities in this area has not been done so far. By introducing an evaluation metric and benchmark set, the paper makes a promising first step in this direction, laying the groundwork for more comprehensive future studies on the role of LLMs in formal specification generation.

The paper is well written and easy to follow. Its language and structure are clear. Only a few minor aspects of the presentation could be improved:
- In several places, a whitespace is missing between the text and citations, particularly in the introduction.
- The font size in most figures is too small, forcing readers to zoom in frequently and disrupting the reading flow.
- The first column of Section 3 is difficult to follow because the individual research questions are introduced only after they have already been discussed at a high level. As a result, the initial discussion lacks context and is hard for readers to understand. Changing the order would resolve this issue.
- There is no reference to HumaneEval. I suppose that the authors refer to OpenAi's dataset with 164 Python programs. My review partly relies on this fact.
- I found Appendices G and H quite insightful and believe that integrating their key takeaways into the main text would strengthen the paper.
- There is a typo in the specification generation prompt in Appendix C.2: "Do not generate any natural language cmments."

The paper clearly positions itself within the wider literature. It also highlights both similarities and differences to existing work, providing a good overview over its core contributions.

To the best of my knowledge, the paper is original. Furthermore, I found many details, such as the question why LLMs fail to prove valid specifications on test cases and the principal barriers to syntactic correctness in Coq, quite insightful and believe that the paper can provide a good foundation for further research in this direction. However, despite the overall positive impression, there are two aspects that raise concerns.

The first aspect concerns the scope of the paper, which is more limited than its title, abstract, and analysis suggests. The paper claims to answer the question of "how powerful are LLMs in generating program specifications?", but the presented approach is limited in two dimensions:
1. the paper only investigates specification in the input language for the Coq interactive proof assistant; and
2. the paper focuses exclusively on Python programs.

Consequently, the paper does not allow drawing general conclusions about the overall capability of LLMs to generate program specifications, but only about the much narrower setting of Coq specifications for Python programs.

To mitigate this issue, the authors could narrow the paper's scope by reformulating the title and highlighting the limitates more prominently throughout the text. Alternatively (and that would be my hope), the authors could extend their investigation to other programming languages and program verifiers. For instance, the [Competition on Software Verification (SV-COMP)](https://sv-comp.sosy-lab.org/) includes a wide variety of program types and specifications that have proven extremely valuable for benchmarking purposes for over a decade. Furthermore, SV-COMP features an active and easy-to-use ecosystem, including tools for interfacing different software verifiers as well as standardized formats for deriving and validating counterexamples to failed verification attempts. While I understand that expanding the paper's scope involves much work, SC-COMP's ecosystem could reduce this effort noticeably.

The second aspect concerns the setup of the evaluation framework COINS as it is not clear what this framework actually measures. The central issue in this context is the lack of a mathematically precise definition of what constitutes a program specification and which formal properties it should satisfy. From what I could glean from the paper, it appears that the program itself would be a suitable specification since it perfectly captures the input-out relation. In that case, the proposed evaluation framework might confound two different tasks: (1) assessing how well LLMs can generate program specifications for Python programs in Coq and (2) assessing how well LLMs can translate Python programs into Coq.

The authors briefly touch upon the above issue in Section 3.5, where they discuss specifications with fixpoint definitions and inductive predicates (the former being executable, while the latter are not). This distinction seems key, and the author could try to disentangle the evaluation by enforcing that an LLM only generates one type of specification.

Another concern relates to the use of LLMs as a probabilistic verifier. First, probabilistic verification is an established field that differs substantially from what the authors describe in Appendix B.1 ([it relates to formal methods for analyzing systems that involve randomness, uncertainty, or stochastic behavior](https://www.sciencedirect.com/science/article/pii/S0890540183710126)). Second, the use of LLMs to judge whether the specification handles negative test cases properly is problematic: the LLM could be wrong, invalidating the results. In particular, using only one specific LLM as a judge could skew the results in favor of the same LLM when evaluating its capabilities for generating the specifications.

Unfortunately, I cannot think of a better solution in the presented approach, given that proving unsatisfiability is often extremely challenging. Furthermore, the authors address this issue comprehensively through several experiments that vary the LLM judge. However, integrating automated verifiers from the SV-COMP ecosystem might mitigate the issue somewhat.

Apart from the above-mentioned issues, the submission is technically sound. Its design choices are clearly described and well justified from a theoretical perspective. The experiments are well-designed and comprehensive. In particular, the authors assess various alternatives in the individual steps of the COINS evaluation framework (e.g., assessing one single LLM as a judge vs. the same LLM as its own judge), ensuring that the design choices are also empirically justified. A minor drawback is that many details of the experimental setup are placed in the appendices, but the main text generally provides sufficient information for understanding. Furthermore, a total number of 165 programs might be a threat to validity.

The code is publicly available in an anonymized repository under an MIT license. However, I could not run the code as it requires access to an OpenAI-compatible LLM API, which is disallowed by my reviewing policy. Therefore, I cannot speak to the reproducibility of the results.

In conclusion, the paper addresses a relevant problem and represents a valuable first step in this research direction. However, the contributions are somewhat limited in scope and might not measure the intended objective, which is not well reflected in the text. Therefore, I cannot recommend acceptance at this point.


## Post-rebuttal comments

I agree that translating a program alone does not solve the task, and have raised my score. Still, presentation issues and the limited scope remain.

---

> ### Author Rebuttal · Authors · 2026-03-31
>
> # Reviewer EP9M
>
> We sincerely thank Reviewer EP9M for the careful review and for pressing us to clarify what COINS measures. For a concrete worked example, please see our response to Reviewer 7Vey on `sort_third`. There, a specification is a logical proposition over the inputs and outputs of a program, typically non-executable, that describes what the program is supposed to do. In contrast, an implementation is a concrete, executable algorithm that shows how the functionality is realized. COINS evaluates a program's specification by formally proving whether it holds for concrete input-output behaviors.
>
> ---
>
> **Core Question 1.** Is the program itself a proper specification? If so, how do you ensure that COINS measures specification generation rather than translation from Python into Rocq?
>
> The program itself is not a proper specification; the two are fundamentally different objects. A specification is a logical proposition over symbolic inputs and outputs: it states the mathematical condition an input-output pair must satisfy, and may include non-executable logical terms such as quantifiers and inductive predicates. An implementation, by contrast, describes how to compute an output. Different implementations (e.g., insertion sort, merge sort, quicksort) may satisfy the same specification (e.g., "the output is a sorted permutation of the input"), but a translation would be tied to one specific algorithm.
>
> COINS enforces this distinction structurally. Our unified specification signature is `f_spec (input : T1) (output : T2) : Prop`, i.e., `T1 -> T2 -> Prop`. A translated program would have type `T1 -> T2`. These are fundamentally different Rocq types; one cannot be mistaken for the other. Furthermore, the LLM is explicitly prompted to generate a `Prop` predicate, not to translate the Python code.
>
> Empirically, Section 3.5 provides additional evidence: if models were simply translating Python into Rocq, we would expect near-universal use of executable `Fixpoint` definitions. Instead, ~60% of specifications from both human experts and strong models are entirely non-executable, confirming that the task requires genuine specification reasoning rather than mechanical translation.
>
> ---
>
> **Additional Questions**
>
> *"Probabilistic verifier."* We acknowledge that this term is misleading; we simply meant that an LLM may fail to prove a true theorem. We will replace it in the revision. Once a proof is accepted by Rocq, the result is reliable by soundness. The harder issue arises in the negative stage: directly asking the LLM to prove a negative-case theorem false is extremely difficult. In pilot attempts, the LLM often behaved unstably, explaining unprovability or rewriting the theorem. To stabilize the pipeline, we first let the LLM filter out obviously incorrect theorems, then attempt proof construction; if a negative-case theorem can be proved, the specification fails to reject it. Any specification counted as passing does so only through proofs checked by Rocq; the LLM does not judge correctness, Rocq makes the final decision.
>
> *Scope.* The current work studies LLM generation of formal Rocq specifications for HumanEval problems, where the Python code serves as a reference implementation rather than the core object of study. A more precise title would be: *"How Powerful Are LLMs in Generating Formal Program Specifications?"*
>
> *SV-COMP and scale.* We chose HumanEval for its representativeness and rich test suite (HumanEval+) and appreciate the suggestion to consider SV-COMP in future extensions. But compared with the SV-COMP subsets used in prior works such as AutoSpec (CAV 2025), HumanEval should not be viewed as a simpler setting. To the best of our knowledge, pushing HumanEval problems all the way to formal specifications and proofs remains very challenging, and there is no evidence of complete formal verification over the full benchmark. CLEVER's end-to-end Lean verification achieves only 0.621%, with equivalence proofs at 1.863% and implementation-satisfies-spec proofs at 8.696%. We will also state the evaluation scale more clearly: while HumanEval contains 164 problems, for each evaluated frontier model COINS generates and checks about 100,000 instantiated theorems and proofs. More details about dataset choice can be viewed in our response to Reviewer HuBT.
>
> *Presentation.* We thank the reviewer for the detailed feedback. We will: (1) fix missing whitespace before citations; (2) increase font sizes in all figures; (3) reorder Section 3 to introduce each RQ before its discussion; (4) add a proper citation for HumanEval; (5) incorporate key findings from Appendices G and H into the main text; (6) correct the typo "cmments" in Appendix C.2.

---

> > ### Author Rebuttal · Reviewer_EP9M · 2026-04-01
> >
> > The author's answered my questions satisfactorily. Given the additional context and example in another review, I agree that translating a program alone does not solve the task. In that light, I will change my score. Still, multiple presentation issues remain, and the scope is still somewhat limited.

---

> > > ### Author Response · Authors · 2026-04-01
> > >
> > > We sincerely thank the reviewer for the thoughtful follow-up and for reconsidering the score. We are glad that our clarification and example helped resolve the concern. We also appreciate the remaining suggestions on presentation and scope, and will use them to further improve the paper.

---

### Official Review · Reviewer_HuBT · 2026-03-10

**Soundness:** 4
**Presentation:** 4
**Significance:** 4
**Originality:** 3
**Overall Recommendation:** 5
**Confidence:** 4

**Summary:**

COINS is a framework for assessing the quality of LLM-generated formal software specifications. COINS is based on Coq, a programmatic proof assistant programming language.
To ground the validity of COINS, the authors provide evaluation results across frontier models and perform all comparisons with a curated dataset of expert-written Coq specifications for 164 HumanEval problems.

**Compliance With Llm Reviewing Policy:**

Affirmed.

**Final Justification:**

My initial review commended the in-depth investigation of using LLMs to generate program specifications. The rebuttal clarified lingering concerns about the "experts" who wrote the ground-truth specifications, and I think the findings of this paper would be a welcome contribution to researchers and practitioners interested in using LLMs for software formal verification.

**Key Questions For Authors:**

* L263 points out that there’s convergence between REJECT_ALL and PASS_ALL likely due in large part to the simplicity of HumanEvan code. Any thoughts on how your findings will extend to harder code?
* Can we get some background on the experts who wrote the human-written Coq proofs? Some transparency into their quality, similar to that of inter-annotator agreement or something, would increase the value of that artifact as a contribution to the community.

**Limitations:**

Yes

**Strengths And Weaknesses:**

## Strengths
* The paper is well-written and motivates an interesting problem, especially in the modern day of increasingly LLM-written software.
* Results are isolated into investigated research questions. Each of these sections details why the question matters, why it is difficult to measure, and how the authors design their experiments to address the challenge and answer the question to the best of their ability. The findings, including a clean error taxonomy across several different models, robustly support the thesis that frontier LLMs are still weak in writing formal specs.
* The authors open-source their code and provide the human-written dataset. This increases transparency and offers value to other researchers who may be interested in using elements of COINS or the dataset.
* The appendix includes extensive qualitative analysis to provide further backing and insights into the claims made int he same body.

## Weaknesses
* HumanEval is a somewhat old benchmark, and the LLM coding community has largely moved on to harder benchmarks. Given that HumanEval examples are somewhat simple (as pointed out in L263) and have extensive input-output suites, it would have strengthened the paper to include results on a benchmark that is more realistic to modern software development challenges. However, I recognize the cost of asking experts to write ground-truth specifications, and still believe the results are valuable given how poorly the frontier models perform with such a "simple" dataset, so this is not a weakness I am particularly concerned about.

---

> ### Author Rebuttal · Authors · 2026-03-31
>
> # Reviewer HuBT
>
> We are grateful to Reviewer HuBT for the strong support and for recognizing the motivation, empirical structure, and artifact value of the paper. We especially appreciate the constructive questions on how the findings may extend and on how to better document the human-written specifications.
>
> ---
>
> **Core Question 1.** L263 suggests the convergence between REJECT_ALL and PASS_ALL may be due to the simplicity of HumanEval. How might the findings extend to harder code?
>
> On harder code, we expect REJECT_all to become more informative rather than less: more complex programs have a larger space of over-approximation, so the gap between accepting positives and rejecting negatives should widen, making the convergence less likely to hold.
>
> ---
>
> **Core Question 2.** Can the authors provide background on the experts who wrote the human-written Rocq proofs/specifications, and stronger transparency on artifact quality?
>
> The specifications were written and cross-reviewed by researchers with formal verification backgrounds, all of whom have multiple years of theorem-proving experience and substantial prior experience reading and writing Rocq code. We will add a detailed description of the annotators' qualifications and the cross-review process in the revision.
>
> ---
>
> **Additional Questions**
>
> *Dataset Selection.*  We understand the reviewer's concern about the difficulty of HumanEval and our choice to focus on it in this work. We selected HumanEval because it exhibits diverse program constructs, captures representative verification tasks, and is a highly influential benchmark. We did not include more datasets mainly due to the substantial human effort needed to construct high-quality formal specifications and the heavy experimental cost of evaluating them. Even so, our study already involves more than 100,000 theorem-proving attempts, so the evaluation is still substantial in scale.
>
> While HumanEval is considered simple for code generation, it still captures many canonical challenges in formal verification, including string induction, list recursion, arithmetic boundary reasoning, and reasoning over nested data structures. For instance, specifying string properties in Rocq requires precise recursive predicates over inductive strings, while permutation-related tasks rely on `Permutation` predicates from Rocq's standard library, both of which are well-known difficulties in verification.
>
> More importantly, "simple for code generation" does not mean "simple for formal specification generation." Appendix G shows that weak models achieve only 4.1% valid-spec success on string problems, and even strong models cover only 40.8% of string tasks, far below their near-saturated coding performance. The fact that the best model reaches only 28.05% REJECT_all on these programs further shows that formal specification remains genuinely difficult.
>
> Finally, COINS is not specific to HumanEval. Whenever test cases are available, COINS can be used to evaluate formal specifications, which makes the methodology practically scalable.

---

> > ### Author Rebuttal · Reviewer_HuBT · 2026-04-03
> >
> > Thank you for the clarifications. I maintain my score, and look forward to follow-up work that investigates extensions and expanded application of COINS.

---

> > > ### Author Response · Authors · 2026-04-03
> > >
> > > We sincerely thank the reviewer for the thoughtful follow-up and for recognizing that our rebuttal has adequately addressed the concerns. We greatly appreciate your acknowledgment and encouragement.
> > >
> > > We also agree that extending COINS to broader settings is an important direction. As future work, we plan to build on the COINS evaluation framework toward fully automated C program verification with code agents, which we believe is a promising next step for scaling formal specification and verification pipelines.
> > >
> > > Thank you again for your valuable feedback and support.

---

### Official Review · Reviewer_7Vey · 2026-03-11

**Soundness:** 2
**Presentation:** 2
**Significance:** 2
**Originality:** 2
**Overall Recommendation:** 2
**Confidence:** 4

**Summary:**

This paper presents COINS, a framework for evaluating the power of LLMs in generating program specifications.  When
using a proof assistant (e.g., Rocq or Lean) developers write specifications and proofs for those specifications.  There
has been substantial work on completing proofs, but limited work on generating specification and evaluating the power of
LLMs in generating such specifications (from natural language).  COINS assesses the quality of generated specifications
using test cases.  The paper presents evaluation of popular LLMs using COINS and concludes, besides other findings, that
LLMs are performing poorly on generating program specifications.

**Compliance With Llm Reviewing Policy:**

Affirmed.

**Key Questions For Authors:**

How do you envision COINS being used in the future and what should be the next focus of developers and LLM developers?
What is the relation of your work to property based testing?
What is the relation of your work to prior work on mutation testing?
Why set the temperature to 0.7?

**Limitations:**

The authors have not addressed the limitation of the approach.
Please see the reviews for details.

**Strengths And Weaknesses:**

**Strengths and Weaknesses**

- *Soundness*:
  - Strengths:
    - Program specification (and use of proof assistants) is becoming more prevalent, thus this paper belongs to an important space
  - Weaknesses:
    - Although the paper sells COINS as a framework, it would be more appropriate to call it "methodology".
    - It is unclear how COINS can be used going forward and how it should be expanded.
    - Despite the expectation (by this reviewer) COINS requires substantial human effort during the entire process
    - There has been substantial work on property based testing used for specification checks (in Rocq); this work was largely ignored in this paper
    - There has been some work on mutation analysis (for Coq/Rocq and Lean); this work was largely ignored in this paper

- *Presentation*
  - Strengths:
    - Tables and figures were mostly clear and well presented
  - Weaknesses:
    - Overall, accomplished work is not well presented: the paper never introduces a single example of a specification, proof objects, test case (neither positive nor negatives).  The authors either assumed that everyone is familiar with these terms (incorrect) or that ICML audience would not like to know more about them (incorrect)
    - Coq-based = Coq was renamed to Rocq some time ago, and it would be appropriate to use the latest name.
    - theorem proving(Xin = missing space before (
    - invariant inference( = lack of space before ( is a global problem in this document
    - are "negative cases" same as "counterexamples"
    - and equivalence verification—by constructing acceptance = introduction talks about negative side of equivalence checking, but then the approach does rely on it
    - These human-authored specifications serve a critical baseline in our framework. = this is the same issue that prior work faces when it comes to eval: it requires substantial expertise and effort.
    - On average, each specification is evaluated against several thousand positive test cases per problem. = if average is 755, how is that "several thousand"?
    - We fix the temperature to 0.7 = why?

- *Significance*
  - Strengths:
    - As already stated, work related to proof assistants and verification is very much relevant nowadays
    - Understanding the limitations of LLMs can help us improve LLMs, humans (and potentially description written in natural language, which was never discussed in the paper by the way)
  - Weaknesses:
    - It was unclear how the authors intend others to use COINS going forward. Is this supposed to be one of "benchmarks"?  Further discussion on the impact of the findings, with respect to developers, tool developers, would be beneficial

- *Originality*
  - Strengths:
    - Using tests appears to be the key claim in the paper about the novelty of the evaluation steps
  - Weaknesses:
    - As already stated, property based testing has been used for years
    - Similarly, there are at least 2-3 tools out there for mutation testing in the space of proof assistants (covering that aspect as well)

---

> ### Author Rebuttal · Authors · 2026-03-31
>
> # Reviewer 7Vey
>
> We thank Reviewer 7Vey for the feedback.
>
> We clarify one distinction. COINS is not testing. It verifies whether a specification is satisfied by a program's input-output pairs by constructing formal proofs in Rocq. This is non-trivial because a program describes an executable computation, while a specification describes the logical relation that inputs and outputs must satisfy. In many cases, specifications are not executable: they involve quantifiers, inductive predicates, and complex logical reasoning. We illustrate with an example.
>
> In `sort_third`, the implementation is an algorithm, while the Rocq specification is a proposition:
>
> ```
> Definition sort_third_spec (input output : list Z) : Prop :=
>   Permutation input output /\
>   (forall(i:nat),(i<length input)%nat->(i mod 3<>0)->
>     nth i output 0%Z = nth i input 0%Z) /\
>   (forall(i j:nat),i<length output/\j<length output/\
>     i mod 3=0/\j mod 3=0/\i<j->
>     (nth i output 0 <= nth j output 0)%Z).
>
> Example sort_third_positive :
>   sort_third_spec [2;1;3;7;8;9;10] [2;1;3;7;8;9;10].
> Proof. unfold sort_third_spec. ... Qed.
>
> Example sort_third_negative :
>   sort_third_spec [2;1;3;7;8;9;10] [7;1;3;2;8;9;10].
> ```
>
> COINS turns concrete input-output pairs and the specification into proof obligations. A positive pair should be provable; an incorrect pair should not. The object being evaluated is the specification itself.
>
> ---
>
> **Core Question 1.** How do you envision COINS being used in the future?
>
> We see COINS as an automated evaluation signal when the reference implementation and trusted behaviors are fixed. For LLM developers, COINS serves as a benchmark for formal specification generation; our multi-stage pipeline provides fine-grained diagnostics beyond pass/fail, including the models' ability to generate formal proofs. For verification practitioners, COINS offers a lightweight way to approximately verify specifications against test suites without full equivalence proofs.
>
> ---
>
> **Core Question 2 & 3.** Relation to property-based testing and mutation-testing work?
>
> We will cite these works in the revision. Formal specifications, particularly those using quantifiers and inductive predicates, are purely logical and cannot be executed. Evaluating them requires proof construction rather than testing, which is the gap COINS fills.
>
> QuickChick (Dénès et al.) can only test executable Rocq properties by generating random inputs, which does not apply to non-executable specifications. We provide two versions of our specification suite, one permitting `Fixpoint` and one using only `Inductive` predicates; even in the permitting version, ~60% are entirely non-executable. COINS takes trusted test cases and uses proof construction. The trust anchors differ: QuickChick trusts the property and questions the implementation; COINS trusts the test cases and questions the specification.
>
> Prior mutation-analysis tools such as mCoq and MutantChick mutate Rocq artifacts (`Definitions`, `Fixpoints`, `Inductive` types) to study whether existing `Lemmas`/`Theorems` and proofs can distinguish them, i.e., they evaluate the adequacy of the proof suite. COINS uses mutation in the opposite direction: it mutates Python implementations to generate negative test cases, then checks whether the specification rejects these incorrect behaviors, i.e., it evaluates the completeness of the specification.
>
> ---
>
> **Core Question 4.** Why temperature 0.7?
>
> Proof generation goes through multiple rounds of Rocq feedback and revision, benefiting from exploration rather than deterministic decoding. We used 0.7 to balance precision and exploratory search.
>
> ---
>
> **Additional Questions**
>
> *Methodology / human effort.* The human-written references are needed only for our experimental comparison. In practice, evaluating specifications requires an LLM to generate a specification and then construct formal proofs that it holds for trusted test cases; no human effort is involved. We will unify "negative cases"/"counterexamples" to a single term and correct "several thousand" to 755.98 per problem.
>
> *Equivalence checking.* The core COINS pipeline does not rely on equivalence proofs. Its main stages are syntax checking, positive-instance acceptance, and negative-instance rejection. The equivalence analysis is auxiliary: full equivalence proofs are often too difficult and obscure specification quality. The low pass rate shows why COINS is useful, providing a discriminative signal without full equivalence verification.

---

> > ### Author Rebuttal · Reviewer_7Vey · 2026-04-03
> >
> > We thank the authors for responding to our comments.
> >
> > The paper requires substantial revision to address all the comments and integrate feedback into account. At this point, as already described, it lacks examples, proper description of a use case, comparison with prior work (yes related work is something that is similar but not the same). We have not given sufficient information about the dataset, so we do not even know if anyone would care about proofs for the selected set of methods.

---

> > > ### Author Response · Authors · 2026-04-03
> > >
> > > We explain our work in further and provide a response as follows.
> > >
> > > **"The lack of examples and use case description"**
> > >
> > > As explained in our previous response, our primary contribution is an evaluation methodology, which is not driven by specific examples. Due to page limits, we prioritized the framework design, experiments, and main findings in the main content.
> > >
> > > We have provided extensive examples and detailed analysis in the appendix — including ill-formed recursion errors (Figure 8), concrete proof failure modes (Figures 9 and 10), and a full case study of a syntactically valid but semantically incorrect specification (Figure 11). We consider to move several representative examples into the main body in the revision.
> > >
> > > **"Comparison with prior work"**
> > >
> > > As we explained in our previous response, our work proposes a novel evaluation methodology for specification quality in program verification, and to the best of our knowledge, there is no similar work providing method focusing on evaluating the power of LLM in proving if a test case can pass a formal specification.
> > >
> > > This addresses a fundamentally different problem from testing you mentioned, therefore property-based testing tools like QuickChick and mutation analysis tools like mCoq are not directly comparable to our approach. We will add an explanation in the revision. We also note that the other reviewers share this understanding and did not raise missing reference concerns.
> > >
> > > **"Not given sufficient information about the dataset, so we do not even know if anyone would care about proofs for the selected set of methods"**
> > >
> > > For dataset, HumanEval is one of the most widely used benchmarks in the code generation community, we did not include detailed introduction, statistics and examples of the HumanEval dataset in the main paper. Our test suite construction is described in detail in Section 2.1, with full algorithmic details in Appendix A. Our rationale for dataset selection is discussed in our responses to reviewers HuBt and EP9M.
> > >
> > > For *"we do not even know if anyone would care about proofs for the selected set of methods"*, we would like to point out that LLM-enhanced formal program verification is an active research direction, which attracts researchers in AI, software engineering, formal methods and programming language, since LLMs bring new hope to solve this classic problem. For instance, CLEVER (Thakur et al., 2025), discussed extensively in our paper, constructs formal specifications and equivalence proofs for HumanEval in Lean.

---

### Official Review · Reviewer_esdb · 2026-03-18

**Soundness:** 3
**Presentation:** 2
**Significance:** 3
**Originality:** 3
**Overall Recommendation:** 4
**Confidence:** 3

**Summary:**

The paper investigates the capabilities of Large Language Models (LLMs) in generating formal program specifications. The authors identify a critical bottleneck in current evaluation methodologies: existing benchmarks either rely on verifying implementation conformance or demand full semantic equivalence proofs, both of which conflate the quality of the generated specification with the inherent difficulty of the formal proving process.
To address this, the authors propose COINS (COq-based INstantiated Specification evaluation), a framework that evaluates Coq specifications by instantiating them on concrete test cases. A specification is deemed high-quality if it can provably accept positive test cases (from HumanEval+) and reject negative test cases (generated via mutation testing). The authors manually construct a ground-truth dataset of human-written Coq specifications for all 164 HumanEval problems to anchor their evaluation. Their large-scale empirical study across six state-of-the-art models reveals that while frontier models approach human-level performance on some tasks, generating precise, syntactically valid formal specifications remains a formidable challenge, with the best model (Gemini 3 Pro Preview) achieving only a 28.05% success rate.

**Compliance With Llm Reviewing Policy:**

Affirmed.

**Key Questions For Authors:**

Mutation testing is used to generate negative examples. How sensitive are the results to the mutation strategy? Have the authors evaluated whether the generated negatives sufficiently cover meaningful semantic deviations?
How would the COINS framework scale to more realistic software verification benchmarks that involve mutable state, loops with invariants, or concurrent behavior?

**Limitations:**

yes

**Strengths And Weaknesses:**

Strength:
The shift from all-or-nothing semantic equivalence checking to test-case-based formal reasoning is an effective solution to the evaluation duality problem. This approach provides a discriminative signal for assessing specification quality in isolation.
The creation of 164 expert-reviewed, human-authored Coq specifications for the HumanEval benchmark provides a baseline for the formal verification and neuro-symbolic communities.
The paper evaluates a diverse set of modern LLMs (including GPT-5, Claude 4.5 Opus, Gemini 3 Pro Preview, and DeepSeek-V3.1), providing  up-to-date frontier capabilities.

Weaknesses
The evaluation is restricted to HumanEval, which consists mostly of relatively simple algorithmic tasks with straightforward functional input–output relations. While useful as a controlled benchmark, it does not reflect the complexity of real-world specifications involving rich invariants, state transitions, concurrency, or system-level behavior.
The negative test cases are generated via AST mutations of canonical implementations. This approach may fail to capture deeper semantic deviations or adversarial edge cases, particularly for more complex specifications.

---

> ### Author Rebuttal · Authors · 2026-03-31
>
> # Reviewer esdb
>
> We sincerely thank Reviewer esdb for the positive assessment and for highlighting both the methodological contribution and the value of the human-written specification dataset.
>
> ---
>
> **Core Question 1.** Mutation testing is used to generate negative examples. How sensitive are the results to the mutation strategy, and do the negatives cover meaningful semantic deviations?
>
> Our goal in the negative stage is to generate behaviors that are genuinely hard to distinguish from correct ones, rather than trivial mismatches. Empirically, the results are not sensitive to the mutation strategy: in supplementary experiments, we compared our AST-level mutations with simpler synthetic strategies (e.g., directly perturbing input-output pairs) and observed no significant difference in discrimination. A plausible reason is that specifications passing all positive cases already tend to be tight enough to reject most negatives, as reflected in Table 2 where REJECT_all and PASS_all values are nearly identical across all models.
>
> Regarding coverage, although completeness cannot be fully quantified, our two-stage design (AST-based mutations + LLM-guided semantic mutations, detailed in Appendix A) is intended to cover both structural and semantic deviations. In supplementary experiments on full negative cases, our mutation pool produces 620 mutants and 271,701 negative cases in total. The human-written references kill 572/620 mutants, while the strongest model (Gemini 3 Pro Preview) kills 485/620, suggesting the negatives do capture meaningful behavioral differences rather than trivial ones.
>
> ---
>
> **Core Question 2.** How can COINS extend to realistic software verification benchmarks involving mutable state, loop invariants, or concurrency?
>
> We appreciate this forward-looking question. The central principle of COINS, using instantiated proof obligations to evaluate candidate specifications, is not tied to HumanEval specifically. HumanEval includes programs with mutable state (e.g., lists), and COINS can handle such stateful programs. Loop invariants are themselves specifications of intermediate program states; as long as appropriate intermediate test cases are provided, evaluating them follows the same framework. Concurrency is not yet addressed in our current work, but is not a limitation of COINS itself: it depends on the expressive power of the underlying specification language and proof assistant (e.g., concurrent separation logic in Rocq).
>
> We will make this more explicit in the revision and position COINS as an evaluation methodology that can be ported to richer verification settings in future work.

---

> > ### Author Rebuttal · Reviewer_esdb · 2026-04-03
> >
> > My concerns are fully resolved.

---

> > > ### Author Response · Authors · 2026-04-03
> > >
> > > Thank you very much for your acknowledgment and for the positive feedback. We truly appreciate the time and effort you dedicated to reviewing our work.

---

### Decision · Program_Chairs · 2026-04-30

**Decision:**

Accept (regular)

**Comment:**

The paper introduces COINS, a Coq-based framework for evaluating LLM-generated formal program specifications. Instead of requiring full semantic equivalence proofs or only checking whether an implementation satisfies a specification, COINS evaluates generated specifications by instantiating them on concrete input-output behaviors and constructing Coq-checked proof obligations. The paper also contributes a curated set of human-written Coq specifications for all 164 HumanEval problems and uses this framework to compare several recent LLMs.

Reviewers generally found the problem important and timely. They appreciated the focus on specification generation rather than proof completion, the attempt to disentangle specification quality from prover limitations, the expert-written reference specifications, and the diagnostic empirical results showing that even strong models still struggle to generate precise Coq specifications. The rebuttal also clarified several technical points, including the distinction between specification evaluation and executable testing, the role of positive and negative test cases, the relation to property-based testing and mutation testing, the use of HumanEval, and the concern that the task might collapse into translating Python programs into Coq.

The main remaining concerns are about scope, positioning, and presentation. The evaluation is limited to Coq/Rocq specifications for Python HumanEval programs, so the title and claims should be narrowed to avoid implying a general answer for all program specification settings. HumanEval is useful as a controlled benchmark, but it does not cover richer verification settings such as system-level invariants, concurrency, or more realistic software verification benchmarks. Reviewers also raised valid concerns about the need for clearer examples, clearer use cases, better integration of appendix case studies into the main text, and a fuller discussion of related work on property-based testing and mutation analysis. One reviewer remained negative after rebuttal and felt that these issues require substantial revision.

The concerns above are real and should be addressed in the camera-ready version, but I view them primarily as limitations in framing and scope rather than fatal flaws in the core contribution. The paper provides a useful and reasonably well-supported evaluation methodology for a difficult and underexplored problem, and three of the four reviewers were ultimately positive after rebuttal. Therefore I recommend weak acceptance.